# A zero-cost attention-based approach to promote cleaner streets: A Signal Detection Theory approach in Parisian streets

**Rita Abdel Sater** [ID]*, **Mathilde Mus, Valentin Wyart, Coralie Chevallier**

Département d'études Cognitives, LNC2, Ecole Normale Supérieure, INSERM, Université PSL, Paris, France

* rita.a.sater@gmail.com

## Abstract

In an effort to inform interventions targeting littering behaviour, we estimate how much a change in trash-bag colour increases trash can visibility in Paris. To that end, we applied standard Signal Detection techniques to test how much changing trash-bag colour affects subjects' trash can detection rates. In three pre-registered studies, we found that changing trash bag colour from grey to either red, green or blue considerably increases the perception of bins in British (tourist) and Parisian (resident) samples. We found that changing the bag colour from grey to blue increased visibility the most.

## Introduction

Littering and improper waste disposal in public spaces is an important challenge for communities and local authorities. Maintaining a clean public environment is actually often reported as a top priority in large and touristic cities. In Paris' latest participatory budget poll, Parisians allocated the highest number of votes to a project aiming to improve "the living environment through more efficient cleaning of the city" [1]. This has put pressure on the municipality to increase an already sizeable yearly budget of€600 million devoted to keep the city clean and maximise the effectiveness of strategies that are already in place to prevent littering behaviour [2].

Littering is a social problem that not only creates aesthetic damages and exhausts a substantial share of public funds, but it also carries important environmental, physical, and psychological costs [3–6]. For instance, cigarette butts which are by far the most littered item around the globe–with approximately 15 billion cigarettes improperly discarded in nature every day- are a major source of land and aquatic pollution that can have dramatic toxic effects on entire ecosystems [7, 8]. The presence of litter in the urban environment can also deepen existing economic and health inequalities by depressing local investment or discouraging outdoor physical activity in highly littered neighbourhoods [6, 9]. Experimental evidence also confirms that even small amounts of litter can induce a rise of antisocial behaviours ranging from further degradation of the living environment to more serious crimes, such as theft [10].

While some theories suggest that an increase in the number of bins might increase litter in the presence of litter in the environment [11], most empirical studies that have focused on understanding and preventing the widespread prevalence of littering as early as in the 1970s

**Funding:** This research was supported by the Agence Nationale de la Recherche (EUR FrontCog ANR-17-EURE-0017*) to Coralie Chevallier. It was also made possible by the support and funding of the Paris City Hall Green Spaces and Environment Department (DEVE). The funders had no role in study design, data collection and analysis, decision to publish, or preparation of the manuscript.

**Competing interests:** The authors have declared that no competing interests exist.

have confirmed that sites with more receptacles tend to have lower littering rates [12–15]. A multilevel analysis identifying the predictors of littering shows that, after controlling for a large number of individual level variables, 15% of littering is still due to some aspect of the physical surroundings, such as the presence or absence of waste receptacles and how conveniently placed they are [13]. A large increase in the number of trash bins therefore has the potential to trigger notable adjustments in behaviour by removing seemingly small barriers hindering proper waste disposal. In an effort to decrease litter in the streets of Paris, 30,000 bins were deployed by the local authorities in 2013. However, one issue with this policy is that increasing the number of bins is costly due to the cost of the bin itself, its installation and the added resources needed to collect trash from additional locations. In Paris, the 30,000 added bins in 2013 cost €2 million, not including collection costs [16]. Despite the important effort, this policy was not sufficient to solve the littering problem. Auditing authorities then recommended yet another increase in the number of bins, raising concerns about the congestion of public space [17].

In this study, we ask whether an increase in the visual saliency of bins can increase the perceived density of bins in a city. Making trash containers more attractive or noticeable has been suggested as an alternative to adding bins in a number of studies [12, 18, 19].

In line with research on nudges, such simple changes to the physical surroundings can be effective in significantly changing behaviour [20]. For example, introducing the scent of lemon in train carriages led to a decrease in litter, suggesting commuters were primed with clean-environment cues [21]. Similarly, a number of studies showed that positioning healthy foods and snacks at eye level more often draws attention to that option and ultimately increases healthy food consumption [22]. Current evidence shows that a more visible bin is more used, regardless of its positioning, which suggests that increasing saliency can be an effective policy to decrease littering [19, 23, 24]. This solution appears particularly relevant in cities, where policy makers strive to keep streets charming and deliberately favour designs that make bins blend with the surrounding urban environment. In Paris, for instance, public trash bins are made of a light grey metal and fitted with discreet grey bags, which means that they are essentially designed to be invisible (Fig 1).

Scientists studying vision have long noted that attention can be modulated in two-ways: endogenously and exogenously [25]. Endogenous attention relies on *top-down*, goal-driven mechanisms, leading the individual to voluntarily seek out information in the environment. Exogenous attention is stimulus-driven and refers to processes that automatically lead salient stimuli to attract attention, through *bottom-up* mechanisms. Exogenous attention is also referred to as bottom-up salience and typically emerges from contrast differences between objects and their surroundings; for example, if the colour of an object is particularly salient compared to the background against which it stands. Multiple studies have indeed shown that salient colours elicit rapid and involuntary attention shifts to the object, regardless of whether subjects had an internally generated (endogenous) motivation to orient their attention to that specific object [26]. Interventions in litter control, such as ex-ante campaigns reminding individuals of littering costs or highlighting the social norm or high fines that punish uncivil behaviour ex-post, might increase endogenous attention and trigger top-down visual search to find the closest bin [27, 28]. Interventions aiming at increasing the salience of bins relative to the surroundings will, on the other hand, trigger automatic attentional capture to the bin. Such an intervention could take the form of a simple change of trash bag colour from grey to more salient colours, leading to an increase in the perceived number of bins.

Using a signal detection task applied to modified photographs of Parisian streets, we ran three studies to measure the impact of simple changes in trash-bag colour on bin detection. Studies 1 and 2 examined the change in trash-bag colour from grey to either red, green or blue

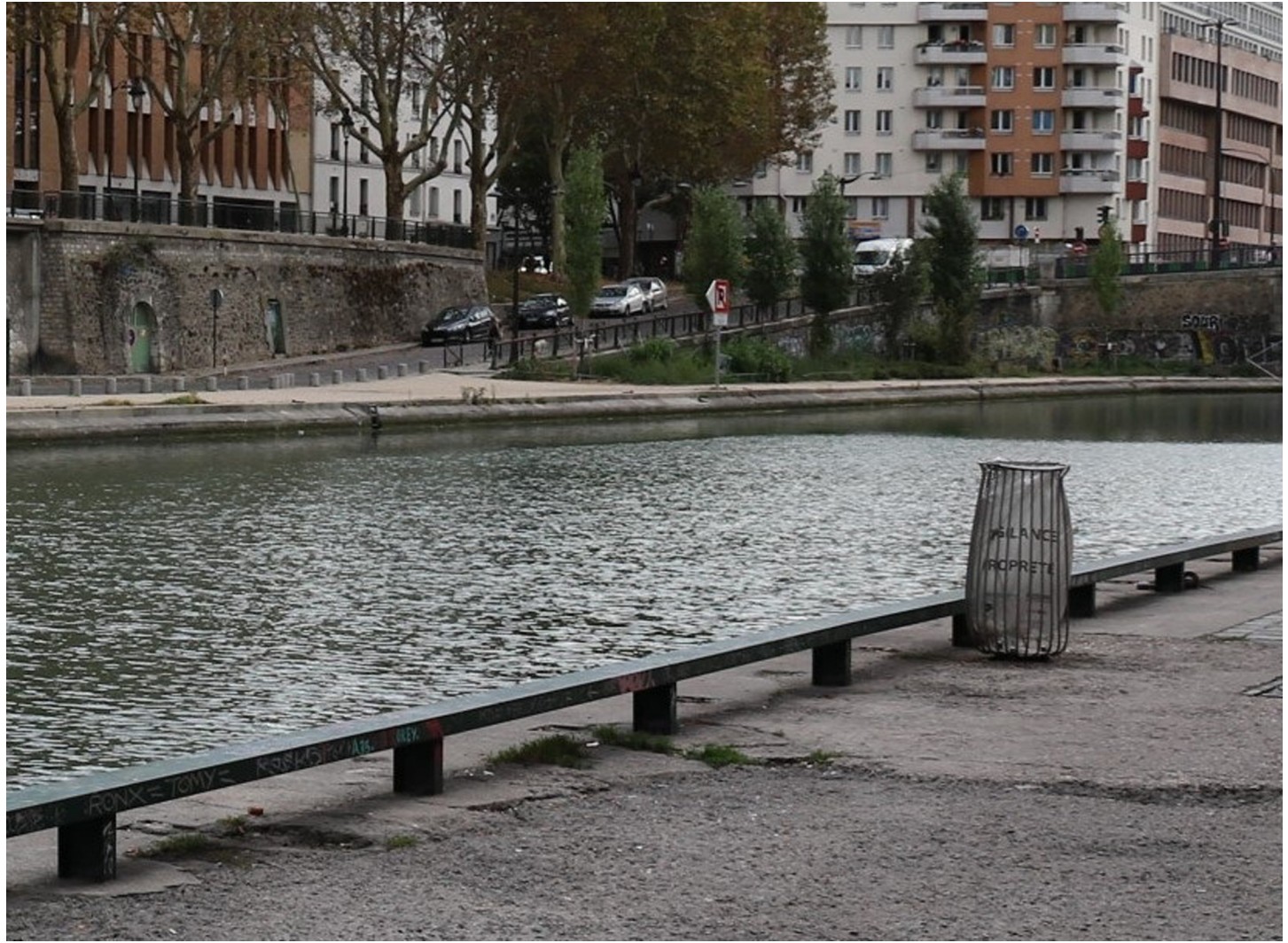

**Fig 1. An example of one of the 30,000 trash bins deployed in Paris since 2013, designed to blend in with the urban space.**

on a British sample and Study 3 examined the change from grey to blue, this time on a sample of Paris residents.

## Study 1

In order to quantify the increase in detectability of trash bins following a change in bag colour from grey to red, a colour that has been reported to be the most salient in a number of vision studies [29, 30], we ask participants to detect the presence or absence of a bin in photographs taken in the streets of Paris, with half of them containing a bin and the other half not containing a bin. We then measure the difference in detection accuracy between the two trash bag colour conditions, using Signal Detection Theory parameters.

### Methods

**Material.** *Photographs*. The visibility of a colour signal in a given environment depends greatly on characteristics of that environment such as the background colour, the

brightness and amount of ambient illumination [31]. 50 unique photos in the initial grey bin condition were taken in Parisian streets to span a variety of settings (brightness, contrast, number of distractors, positioning of the bin, etc.). In order to select these 50 photos we first ran a pilot study on 100 participants with a set of 100 photos, only testing the grey condition. This allowed us to calculate a measure of detectability $d'$, which corresponds to participants' ability to detect the bin (see section "Analysis" below for a more detailed explanation) for each photo in the baseline condition and pick the 50 photos exhibiting bin detectability that were neither too low nor too high. We set the display time to 700ms after noticing that the task appeared to be very hard (accuracy of detection was really low for most participants) for a shorter display time of 500ms.

In order to ensure comparability between the different conditions (grey-bag condition, red-bag condition, no-bin condition) and eliminate potential biases related to the object's visibility, we created three versions of each photograph using Adobe Photoshop: 1) a grey-bag version with the original photograph of the bin fitted with its original grey bag (grey bag condition) 2) a red-bag version with the same bin edited to be fitted with a red bag (red bag condition) 3) and a no-bin version with the same photograph edited to have no bin (no-bin condition). When producing the different versions of the stimuli, we altered the hue of the bag but did not change the contrast. Fig 2 (Panels a-c) shows an example of the 3 versions of the same photograph.

*Questionnaires*. We also collected information on socio-demographic variables (age, sex, income level and educational attainment) and included standard questions used in visual tasks (self-reported colour blindness, self-reported use of glasses or contacts) to assess our sample's representativeness.

**Design and procedure.** The pre-registered experimental task was hosted on the online research platform Prolific Academic and programmed using JavaScript in Qualtrics. Participants were asked to use a full-screen mode on a computer to complete the task, which lasted between 8 and 15 minutes. The same image was seen 4 times by each participant; once in the grey-bag condition, once in the red-bag condition and twice in the no-bin condition. Each participant therefore saw the 200 photographs in a random order. Of these 200 stimuli, 100 contained a trash bin (50 in the red-bag condition and 50 in the grey-bag condition) and 100 contained no bin (each of the 50 photographs containing no bin was presented twice). The 200 trials were split into three blocks separated by two breaks. Both the order of the blocks and the order of the pictures within each block were randomised. In each trial, the picture was flashed in the centre of the screen for 700ms. We used a short display time rather than a display time allowing for visual search because our goal was to focus on automatic and rapid attentional processes [32, 33]. Subjects in a visual search task make multiple fixations and saccades, with each fixation lasting between 200 and 500ms approximately [34]. This means that the chosen presentation time of 700ms gives our participants enough time to perform only two to three fixations. Given the complexity of the visual scenes participants were presented with, participants did not have enough time to fully scan the scene in a top-down fashion.

Participants were then asked to indicate whether they saw a bin or not at the end of each trial, using letters on their keyboard. Both responses and reaction times were recorded. In addition, 20 catch trials were included to screen out participants who were not paying attention during the task. In these trials, participants saw photographs with easily detectable bins that were displayed long enough (1 second) to make the answer trivial. Questionnaire data was collected at the end of the experiment.

**Participants.** Keeping in mind that tourism has a strong impact on littering [35], we focused on a non-Parisian sample, i.e. people who are not familiar with the format of bins in Paris. Considering that the highest number of tourists in the city comes from the United

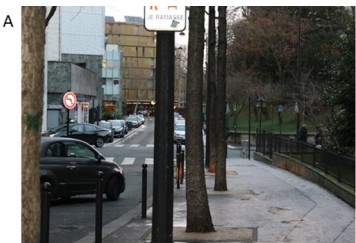

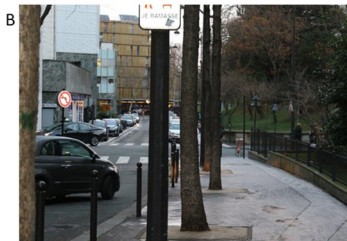 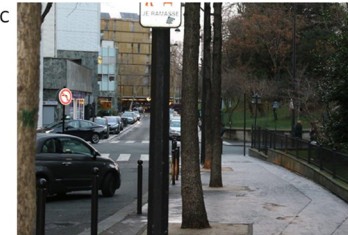

**Fig 2. Example of a displayed image in the three possible conditions.** A. Original image, grey-bag condition. B. Photoshopped image, red-bag condition. C. Photoshopped image, no-bin condition.

Kingdom [36], we recruited 324 British participants through the online platform Prolific Academic. This sample is large enough to detect a small effect size of $dz = 0.15$ ($\alpha$ = 5%, 1-$\beta$ = 80%) following a power analysis with G*Power (version 3.1.9.3) for a two-tailed t-test for dependent means, using outcome variances computed from the pilot study that included 100 participants. The number of participants recruited on the platform took into account a 10% possible exclusion rate. All participants received a compensation of 10.7 euros per hour. We pre-screened participants using Prolific's approval rating; participants with approval ratings below 95% were screened out. We excluded 15 subjects who performed at or below chance in the catch trials. Trials for which reaction times were too short (150 milliseconds or less) or too long (4,000 milliseconds or more) were also excluded. After making sure that no participant had more than 30% missing trials, 307 participants remained in the final analysis (203 females, 104 males, mean age: 36.96, SD = 12.72 years). 2% of the sample reported anomalous colour vision (participants were asked whether they were colour blind) and 60% of them declared wearing glasses or contact lenses to correct their vision, similar to the percentage in the UK population [37, 38]. The lower rate of reported anomalous colour vision (2% in the sample versus 4% in the population) reflects the overrepresentation of women in our sample: according to the NHS, 8% of men and 0.5% of women are affected by colour blindness in the UK population.

**Analysis.** Our study's analysis plan relied on Signal Detection Theory in order to quantify the change in salience resulting from a change in bag colour. Signal Detection Theory was used to measure participants' ability to discriminate between a target stimulus and irrelevant noise [39]. In Signal Detection Theory, participants' responses are classified into one of the following four categories: hits, misses, false alarms and correct rejections. In the red-bag and grey-bag conditions, a "hit" is recorded when the participant detects the bin and a "miss" is recorded when the participant fails to detect the bin. In the no-bin condition, a "false alarm" is recorded when the participant detects a bin and a "correct reject" is recorded when the participant indicates that there is no bin. In this study, we were particularly interested in capturing changes in bottom-up attention in response to increased trash bag salience. We therefore focused on the discriminability $d'$, a metric that allows us to capture earlier stages of visual

processing such as the sensory encoding of stimuli. $d'$ represents the strength of the signal relative to noise, which corresponds to participants' ability to detect the bin.

A discriminability of 0 means that the signal is not distinguished from noise, while higher $d'$ values represent a higher sensitivity (i.e. a situation where it is easier for people to discriminate the bin in the scene). $d'$ can thus be considered as an index of task difficulty. $d'$ is calculated by subtracting z corrected false alarms from hits: $d' = z(H_c) - z(F_c)$, where $H$ is the hit rate, $F$ the false alarm rate and the subscript $c$ the colour condition (*red* or *grey*) and $z()$ the normal probability curve. The difference in mean discriminability between the two colour conditions $d'_{red} - d'_{grey}$ will thus allow us to quantify the change in bin perception following a change from grey to red.

Given that photographs containing the bin are identical to photographs not containing the bin, it is reasonable to assume equal variance of the noise and signal distribution across conditions.

However, participants exhibit a bias in their strategy to set a threshold over which they make the decision that the stimulus is present. This bias is measured by a second metric in Signal Detection Theory, the criterion, or 'response bias', $C$. It thus reflects a participant's tendency to provide one type of response more frequently than the other. It is calculated as follows: $C_C = -\frac{z(H_C) + z(F_C)}{2}$. A criterion $C$ with a value of $0$ shows that the decision threshold is fixed at a level that generates equivalent rates of false positives and false negatives. C will be positive if there are more hits and false alarms than misses and correct rejections.

Signal Detection Theory thus allowed us to get a measure of discriminability $d'$ while taking into account the response bias $C$ [32, 40]. In the analysis, the false alarm and correct rejects rates calculated from the 100 no-bin photographs were then used in the calculation of both $d'_{red}$ and $d'_{grey}$.

Then, to measure the change in performance in a more intuitive manner, we estimated the area under the received operating characteristic (ROC) of each colour condition using the $d'$ as follows (Macmillan, 1993) [41]: $A_{d'} = \phi\left(\frac{d'}{\sqrt{2}}\right)$, where $\Phi(\cdot)$ corresponds to the cumulative normal distribution. The ROC area can be interpreted as the proportion of times subjects would correctly identify the signal [42]. For example, if the trash bag is easy to see, ROC area will be equal to 1. If the trash bag is virtually impossible to detect the participant must guess and the ROC area will be at level of chance: 0.5.

## Results and discussion

$d'$ was calculated for each individual, in each of the two colour condition. A paired t-test for repeated measures was then used to compare the average discriminability change between the two colour conditions. In line with our hypothesis, $d'$ was 23% higher in the red bag condition ($M = 1.65$, $SD = 0.71$) than in the grey bag condition ($M = 1.34$, $SD = 0.62$), $t(307) = 15.59$, $p < .001$, $dz = 0.88$. The standardized mean difference effect size for within-subjects designs is referred to as Cohen's dz and its formula is based on calculations by Rosenthal (1991): $d_z = \frac{t}{\sqrt{n}}$, , given the direct relationship between the t-value of a paired-samples t-test and Cohen's dz. ROC area calculations show that participants accurately identified the bin in 87.8% of the trials in the presence of a red bag ($A_{d'_{red}} = (0.88)$ and in 82.8% of the trials in the presence of a grey bag ($A_{d'_{grey}} = 0.83\%$). There was a 5 percentage point difference when comparing correct identification of bins with red bags compared to grey bags.

Moreover, mean reaction times to detect the bin were lower in the red-bag condition ($M = 1076.02$, $SD = 283.02$) than in the grey-bag condition ($M = 1113.91$, $SD = 286.32$), $t(307)$

= -8.86, $p < .001$, $dz$ = 0.50. The results were not different according to gender (see regression tables in S1 Material). Both accuracy and speed increased in the colour condition, which is consistent with an exogenous attentional effect. However, it is worth pointing out that the reaction times capture several processes beyond attention, including the selection of motor goals and motor planning, which explains why they are on average above one second even though attentional capture happens much faster [43]. In summary, bins with red bags were more visible than bins with grey bags.

## Study 2: Follow-up study with additional colours

We applied the same tools and methodology to other colours most frequently considered, blue and green [44, 45]. These two colours also bear practical relevance for the current study, since they have been considered by the Parisian municipality as potential alternative trash bag colours. In fact, green bags were used by the city from 2013 up until 2019 and were then replaced by grey bags for aesthetic reasons (Personal communication, the department of cleanliness and sanitation (DPE) of the Paris municipality, 2021).

Using the same experimental methods employed to identify the change in visibility when changing bins from grey to red, Study 2 was run to investigate the effect of a change of colour from grey to green on bin visibility (Study 2.A) or grey to blue (Study 2.B). These two follow-up studies were pre-registered. Fig 3 (Panels a-b) shows an example of the 2 versions of the same photograph.

### Participants

312 British participants were recruited for Study 2.A and 315 for Study 2.B through the online Platform Prolific Academic and received a compensation of 10.7 euros per hour. Participants were pre-screened to have a 95% or above approval rating (i.e., participants with more than 95% of previous submissions on the platform being approved). 7 subjects who performed at or below chance in the catch trials were excluded from the analyses of each study. As in Study 1, trials for which reaction times were too short (150 milliseconds or less) or too long (4,000 milliseconds or more) were also excluded. After making sure that no participant had more than 30% missing trials, 301 participants remained in the final analysis for Study 2.A (174 females, 127 males, mean age: $M$ = 35.77, $SD$ = 12.28 years) and 308 for Study 2.B (202 females, 106 males, mean age: $M$ = 36.71, $SD$ = 12.32 years). 2% of participants reported anomalous colour vision and 57% wore glasses or lenses in Study 2.A while 1.5% were colour blind and 60% wore glasses or lenses in Study 2.B.

### Results and discussion

Paired equal variance two-tailed t-tests for repeated measures revealed that $d'$ was 24% higher in the green-bag condition (Study 2.A, $N$ = 301) ($M$ = 1.73, $SD$ = 0.69) than the one observed in the grey-bag condition ($M$ = 1.40, $SD$ = .58), $t(304)$ = 18.59, $p < .001$, $dz$ = 1. Study 2.B ($N$ = 308) tested the visibility of the blue-bag condition and showed the highest increase in discriminability compared to grey out of the three tested colours; in the blue-bag condition, a paired equal variance two-tailed t-tests for repeated measures $d'$ was 35% higher ($M$ = 1.86, $SD$ = .70) than the one in the grey bag condition ($M$ = 1.38, $SD$ = .62), $t(307)$ = 25.26, $p < .001$, $dz$ = 1.44. The results were not different according to gender (see regression tables in S1 Material). In both studies, mean reaction time was shorter ($p < .001$) in the coloured condition (Study 2.A: green $M$ = 1097.31 ($SD$ = 279.66) vs. grey $M$ = 1141.38 ($SD$ = 274.24) and Study 2. B blue $M$ = 1006.68 ($SD$ = 215.12) vs grey $M$ = 1056.42 ($SD$ = 211.03), which shows again a decrease in reaction times in the colour condition in both studies.

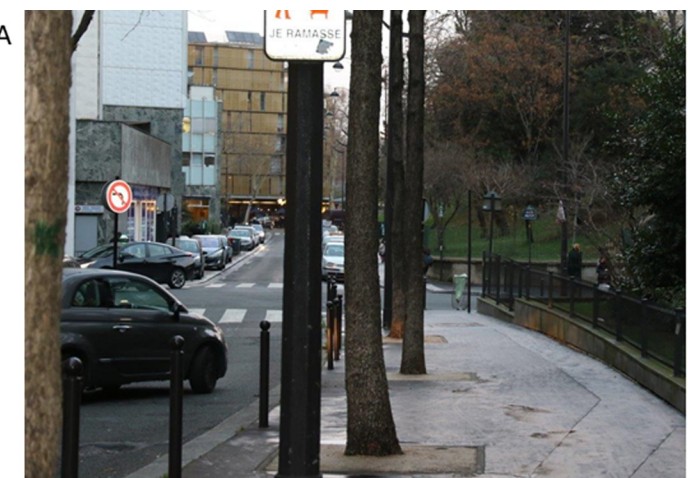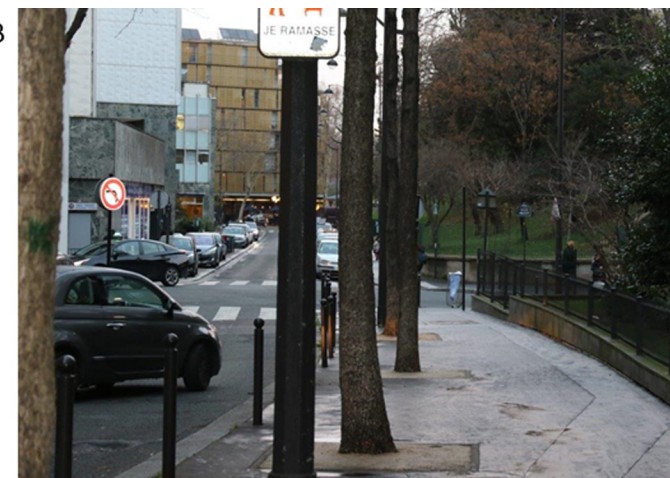

**Fig 3. Example of a displayed image in studies 2.A and 2.B.** A. Modified photograph in the green-bag condition. B. Modified photograph in the blue-bag condition.

We performed a mixed-model ANOVA to compare the results obtained in Studies 1, 2.A and 2.B, using the change from grey to colour as a within subject factor and study number as the between subjects factor, with $d'$ as the dependent variable. The ANOVA assumptions of normality, independence of samples and sphericity were met but variances were heterogeneous. We therefore ran a mixed-model robust ANOVA using the R package WRS2, with 3-level factor STUDY (green, red and blue) and 2-level factor COLOUR (grey, colour). The main effect of COLOUR was significant, $F(1, 912) = 1191.40$, $p < .01$, $\eta^2 = .57$, but the main effect of STUDY was not significant, $F(2, 912) = 2.73$, $p = .07$, $\eta^2 = .01$. We found a significant interaction between STUDY and COLOUR, $F(2, 912) = 22.672$, $p < .01$, $\eta^2 = .047$, which suggests that the effect of the change in colour on $d'$ is different across the studies. An analysis of simple effects showed that the treatment effect was significant for the colour condition (between the different colours), $F(2,912) = 6.40$, $p < 0.01$, but not in the grey condition $F(2, 912) = 0.73$, $p = 0.48$. This suggests that participants responded the same for the grey condition across the three studies, but there was a difference in the coloured trial conditions.

ROC area calculations show that a participant was 6% more likely to correctly identify a bin with the green bag (Study 2.A: $A_{d'_{green}} = 88.92\%$, $A_{d'_{grey}} = 83.81\%$) and 8% more likely to identify a bin with a blue bag (Study 2.B: $A_{d'_{blue}} = 90.52\%$, $A_{d'_{grey}} = 83.57\%$).

We then did three separate mixed ANOVAs to compare the effect size of COLOUR, this time with a 2-level factor STUDY, in order to measure whether there is significant interaction between each pair of studies. An interaction graph (Fig 4) showed the change from grey to colour across all three studies. We found significant COLOUR x STUDY interactions on $d'$ when comparing red (Study 1) to blue (Study 2.B), $F(1, 615) = 35.92$, $p < .001$, $\eta^2 = .05$, and when comparing green (Study 2.A) to blue (Study 2.B) $F(1, 610) = 32.01$ $p < .001$ $\eta^2 = .05$; but no interaction when comparing red (Study 1) to green (Study 2.A), $F(1, 611) = .46$, $p = .45$, $\eta^2 = .00$. We verified that the results were still significant after a Holm-Bonferroni correction for multiple testing.

In summary, Study 2's results showed that changing the colour of bags from grey to green and from grey to blue significantly increased their visibility. A comparison between the three colour conditions showed that a bin with a blue bag was significantly more visible than one with a red or green bag.

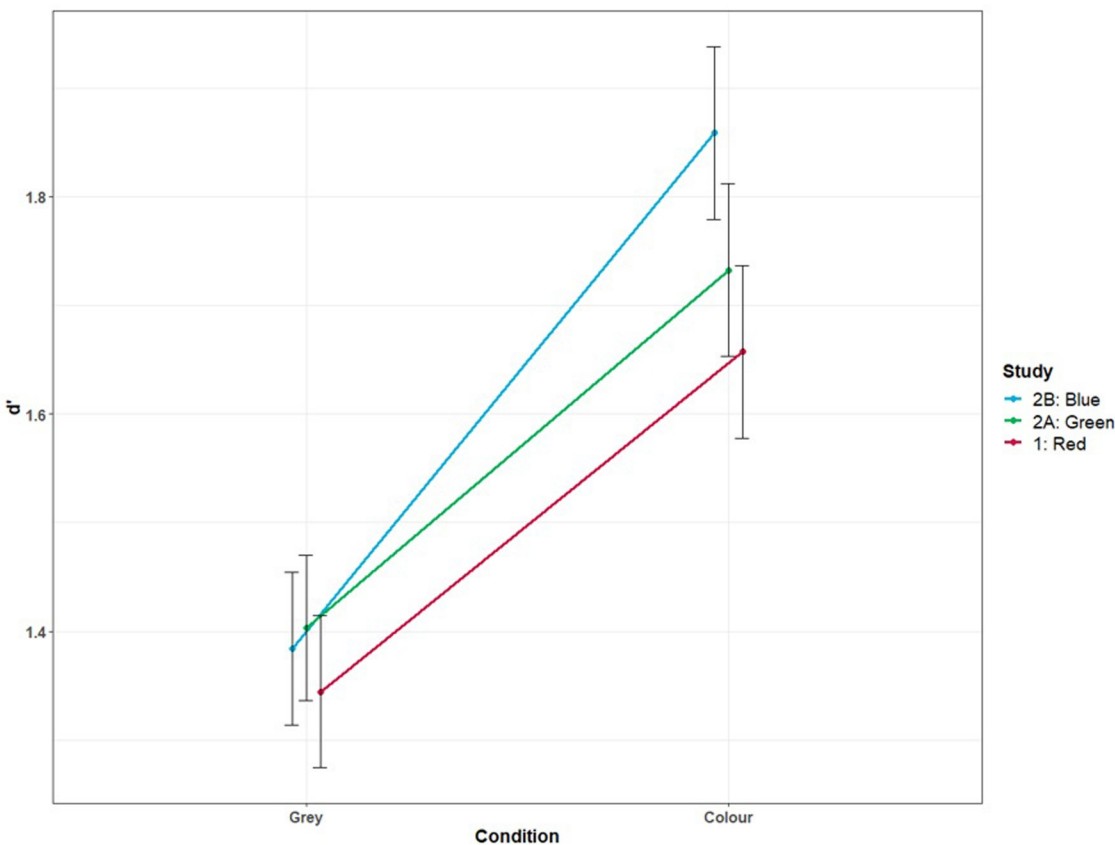

**Fig 4. Interaction graph showing the change in detectability when a bin is coloured, in studies 1 and 2.** Mean and 95% confidence interval.

## Study 3: Study with a Parisian sample

The goal of Study 3 was to test whether the observed changes would hold in a sample of people more familiar with the bins. We replicated the design used in the previous studies in a sample of participants living in the Paris area. The expected effect on a local sample is ambiguous; on the one hand, participants who are familiar with Parisian streets could have a better top-down grasp of the shape and location of the bins, which should make them better at detecting them regardless of colour. This would lead to a more modest or null effect of the colour change. On the other hand, the effect of the novelty of the bin bag colour versus the one the residents are used to, might cause a greater effect on attention. While we purposefully used a non-resident sample in the previous studies to target tourists and disentangle the actual increase in salience from the novelty effect that might wear out fast, it is still crucial to measure the effect on the population most likely to use the bin other than visitors in non-touristic areas of the city. The colour blue was tested against the colour grey, as it was the one that showed the most promising results in Studies 1 and 2.

### Participants

310 participants living in the Paris area were recruited on the platform CrowdPanel and received a compensation of 16 euros per hour. Trials for which reaction times were too short (150 milliseconds or less) or too long (4,000 milliseconds or more) were excluded. After

making sure that no participant had more than 30% missing trials, 283 participants remained in the final analysis (125 females, 158 males, mean age: $M = 39.11$, $SD = 13.84$ years). 2% of the sample reported anomalous colour vision (participants were asked whether they were colour blind) and 61% of them declared wearing glasses or contact lenses to correct their vision.

## Results and discussion

We observed that $d'$ was 40% higher in the blue-bag condition ($M = 2.04$, $SD = 0.85$) than the one observed in the grey-bag condition ($M = 1.46$, $SD = 0.75$), $t(282) = 30.03$, $p < .001$, $dz = 1.74$, using a paired equal variance two-tailed t-tests for repeated measures). A mixed model ANOVA testing 2-level COLOUR factor (grey vs colour) x 2-level STUDY factor (study 2.B vs study 3) confirms that the effect of changing the colour from grey to blue on $d'$ was greater when considering a sample of residents in the Paris Area compared to a British sample; the interaction between STUDY and COLOUR was significant $F(1,589) = 14.91$, $p < .001$, $\eta^2 = .025$. The main effects were also significant; STUDY $F(1,588) = 5.07$, $p = .02$, $\eta^2 = .01$, COLOUR $F(1,588) = 1585.63$, $p < .00$, $\eta^2 = .73$. ROC area calculations show that a participant was 9% more likely to correctly identify a bin with the blue bag ($A_{d'_{blue}} = (92.55\%$, $A_{d'_{grey}} = 84.97\%)$ compared to a bin with a grey bag.

In summary, we observed the same results when we considered a Parisian (local) sample; the visibility of a bin increased when its bag is blue compared to grey. Study 3 also shows that the change in visibility is greater for Parisian participants than it is for British participants.

## Discussion and conclusion

Individuals are often meaningfully influenced by modifications in their environment, even ones that appear minor. One of the main propositions put forward in nudge theory is the importance of alterations to the physical environment, in order to align behaviour with individual and social goals [20]. Understanding how to enhance the visibility of waste bins therefore has the potential to ultimately favour better anti-littering policies with zero-cost nudges. In the context of this study, our exchanges with the Parisian local authorities confirmed that the red, green and blue bags were provided by the same suppliers and cost the same as grey bags, so if the authorities decided to change their bags from grey to red, the cost would be zero. This, however, does not necessarily hold in all contexts where the production of a new coloured bag might incur a relatively low, but non-zero cost.

The results of this study suggest that a mere change in the colour of bins can result in a cost-effective intervention that would increase the visibility of bins. This implies that some previously invisible bins would become visible as a result of the intervention, thereby increasing the perceived density of bins in an urban setting. Interestingly, the increase in *perceived* density is even more dramatic in a follow-up study on a Parisian sample, a population that is already familiar with the bin and its placement. Beyond the samples used in our studies, it would be useful to also test a sample that would be fully representative of the full diversity of passersby in Paris.

This might be particularly relevant for high touristic areas, which suffer from high rates of littering. When considering the installations of more bins in the city, increasing *perceived* bin density, rather than actual bin density, may be a cost-effective policy lever, particularly in cities with an already great amount of bins per citizen, and a persistent littering issue. While changing the colour of bags is a zero-cost intervention, adding bins could have high installation and maintenance costs. Beyond the sizeable monetary costs, changing the colour of bin bags as opposed to increasing their number avoids overcrowding an already congested public space. However, it is important to note that increasing the saliency of bins can entail aesthetic costs

that might be problematic in touristic cities with architectural heritage like Paris, and this will need to be properly evaluated against the costs of high littering.

Our results also provide grounds for future work investigating whether an increased visibility of bins has a subsequent impact on littering behaviour but a number of limitations should be acknowledged. First, our set of experimental online studies may lack ecological validity and looking for a bin in real settings may rely on different mechanisms. Specifically, the short display time in our experiment mimics the impact of increased visibility when people are not actively looking for a bin, which differs from what happens in real-life. Therefore, although the results of our studies provided initial evidence on the superiority of some colours compared to others in increasing salience, a field experiment is needed in order to validate the findings. Moreover, our studies do not allow us to state whether changing bag colour would have an effect on real-life littering. However, prior research provides suggestive evidence that making bins more attractive or noticeable affects behavior [19, 23, 24]. For example, littering rates in a shopping mall were 40% lower around highly noticeable and beautiful trash bins than around unobtrusive ones [23]. In another experiment, a decrease in improper cigarette butt disposal was observed outside a university campus after replacing normal ashtrays by ones that were decorated and eye-catching [46]. Some cities, like Copenhagen and Vienna, have opted for increased salience of bins and have recorded drastic positive results [17].

In all these cited studies, the bin is at the same distance in the eye-catching condition and in the control condition but littering rates are different. This suggests that bottom-up attention plays an important role in littering, perhaps because bottom-up attentional processes affect all passersby in a similar way, unlike less malleable determinants of littering, such as people's individual motivation to look for a bin. Using short display times, as in our studies, is an interesting method to capture such bottom-up mechanisms. More generally, this paper is a telling illustration of how experimental online studies can provide insightful contributions to policy-making and constitute a first crucial step before conducting field experiments. Indeed, if a policy maker wants to increase the *perceived* density of bins, it would be inefficient to simply rely on intuition or even previous laboratory research in different settings; for example, we show that even though red is reported most salient in a number of vision studies [29, 30], it was not as good as blue in increasing the visibility of bins in the Parisian urban setting. It would also be inefficient to go straight to a costly field trial comparing various bin colours. Determining which colour is most likely to be salient in a particular city in an experimental setting first can therefore be a useful evidence-based first step before elaborating a field strategy.

The method we used can be easily taken up by policy makers in order to produce evidence-based decisions on how to increase the visibility of trash receptacles in urban settings. Based on insights obtained in an online experimental setting, decision-makers can then maximise the efficiency of a field intervention, evaluated with rigorous but often costly field trials. For instance, our method could be replicated with a variety of bin shapes and colours to help identify the most effective choice before turning to large-scale implementation or field trials. Well-designed experimental online research can provide crucial insight for policy and is often complementary to field research, but is still greatly underused in the domain of policy development [47].

## Supporting information

**S1 Material.**
(DOCX)

## Acknowledgments

We gratefully thank Arnaud Le Bel Hermile and Mariane Lavallee (Acteurs du Paris durable) for their support and insight throughout this study. This study was supported by the EUR FrontCog grant ANR-17-EURE-0017 and ANR-10-IDEX-0001-02 to PSL.

## Author Contributions

**Conceptualization:** Rita Abdel Sater, Valentin Wyart, Coralie Chevallier.

**Data curation:** Rita Abdel Sater, Mathilde Mus.

**Formal analysis:** Rita Abdel Sater, Valentin Wyart, Coralie Chevallier.

**Funding acquisition:** Rita Abdel Sater.

**Investigation:** Rita Abdel Sater.

**Methodology:** Rita Abdel Sater, Mathilde Mus, Valentin Wyart, Coralie Chevallier.

**Project administration:** Coralie Chevallier.

**Software:** Rita Abdel Sater.

**Supervision:** Valentin Wyart, Coralie Chevallier.

**Visualization:** Rita Abdel Sater.

**Writing – original draft:** Rita Abdel Sater.

**Writing – review & editing:** Mathilde Mus, Valentin Wyart, Coralie Chevallier.

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
