## [Decision Letter · Decision Letter 0]

13 Jul 2021

PONE-D-21-08195

A zero-cost attention-based approach to promote cleaner streets: a Signal Detection Theory approach in Parisian streets

PLOS ONE

Dear Rita Abdel Sater,

thank you for submitting your manuscript to PLOS ONE.  After careful consideration by three experts in the field, we feel that it has merit but does not meet PLOS ONE’s publication criteria as it currently stands.  I appreciate very much the Reviewers' efforts for providing their thoughtful and detailed comments while evaluating the manuscript.  You will see that while mostly finding the research topic exciting (including myself), the Reviewers' opinions have disagreed to a considerable extent.  We invite you, however, to submit a revised version of the manuscript that addresses the points raised during the review process, paying particular attention to the following major points:

1) Please carefully check all technical and statistical issues mentioned by the Reviewers.  Explain if your data sets were tested for normality, and which statistical inference was used for respective data sets.  Also, for each p-value please report which statistic was taken, also considering tests' directionality, multiplicity issues and so forth.

2) Please justify if an ethical approval would (not) be required for your work.  In case of uncertainty, please consult with the PLOS ONE Editorial Team on this issue.

3) Please downscale Discussion so that the conclusions would be better supported by your data. 

Also,  please make sure to thoroughly address all the other Reviewers' comments and suggestions. 

Please submit your revised manuscript within six months from this date as thereafter, any revision has to be considered a new submission.  If you will need more time than this to complete your revisions, please reply to this message or contact the journal office at plosone@plos.org. Please include the following items when submitting your revised manuscript:

We look forward to receiving your revised manuscript.

Thank you for choosing PLOS ONE for communicating your reasearch.

Kind regards,

Sasha

Alexander N. 'Sasha' Sokolov, Ph.D.

Academic Editor

PLOS ONE

Journal Requirements:

We note that you have stated that you will provide repository information for your data at acceptance. Should your manuscript be accepted for publication, we will hold it until you provide the relevant accession numbers or DOIs necessary to access your data. If you wish to make changes to your Data Availability statement, please describe these changes in your cover letter and we will update your Data Availability statement to reflect the information you provide.Please amend either the title on the online submission form (via Edit Submission) or the title in the manuscript so that they are identical.

Thank you for stating the following in the Acknowledgments Section of your manuscript:

This research was supported by the Agence Nationale de la Recherche (EUR

FrontCog ANR-17-EURE-0017*). It was also made possible by the support and funding

of the Paris City Hall Green Spaces and Environment Department (DEVE). We

gratefully thank Arnaud Le Bel Hermile and Mariane Lavallee (Acteurs du Paris

durable) for their support and insight throughout this study.

This research was supported by the Agence Nationale de la Recherche (EUR

FrontCog ANR-17-EURE-0017*) to Coralie Chevallier. It was also made possible by

the support and funding of the Paris City Hall Green Spaces and Environment

Department (DEVE).

The funders had no role in study design, data collection and analysis, decision to

publish, or preparation of the manuscript.

5. We note that Figures 2-3 includes an image of a participant in the study.

Reviewers' comments:

Reviewer's Responses to Questions

**Comments to the Author**

1. Is the manuscript technically sound, and do the data support the conclusions?

Reviewer #1: Partly

Reviewer #2: Yes

Reviewer #3: Partly

2. Has the statistical analysis been performed appropriately and rigorously? 

Reviewer #1: Yes

Reviewer #2: No

Reviewer #3: Yes

3. Have the authors made all data underlying the findings in their manuscript fully available?

Reviewer #1: Yes

Reviewer #2: Yes

Reviewer #3: Yes

4. Is the manuscript presented in an intelligible fashion and written in standard English?

Reviewer #1: Yes

Reviewer #2: No

Reviewer #3: Yes

5. Review Comments to the Author

Reviewer #1: This is an interesting line of research, examining the link between bag color for trash receptacles, and perceptions about the number of receptacles in a public space (what the authors define as bin density). A series of experiments are reported examining the influence of bag color on perceptions of bin density. Results show that red bags produce higher ratings of bin density, compared to grey bags (Study 1). Study 2 shows that green bags also result in higher density ratings, and finally, in Study 3 we see that green bags also work to increase density ratings. Based on these results, the authors argue that switching the bag color to blue can increase bin density, and thereby decrease littering rates across the city of Paris.

While the topic of the study was exciting, the results do not warrant publication in their current form. Listed below are my key concerns, and unfortunately, I do not believe that these issues can be addressed without additional data.

1. The critical missing piece is data showing the connection between bin density and litter. The authors make a fundamental assumption that higher bin density will result in lower littering rates, but this is not substantiated. In fact, from the social psychological research on normative influence, there is reason to predict that increased bin density could spark MORE litter in environments that have a high degree of existing litter (see research by Cialdini on the focus theory of normative conduct).

2. More clarity is needed about the theoretical foundations of this research. The authors briefly mention the Signal Detection Theory, and later 'Nudge Theory', yet the results are not integrated into these two foundations. Specifically, Study 3 revealed the strongest increases in perceptions of receptacles as a result of switching to blue bags, compared to green and red bags. However, the authors do not address that this result is contradictory to the signal detection theory's claims for increased attention to the color red.

In addition, while Nudge Theory is cited, there is a body of research in social psychology around norms that is highlight relevant and should be integrated into the paper.

3. One of the implications of this research is support for supplementing the cost associated with increased trash bins compared with increased salience of existing bins. However, the authors mention that the city previously utilized green bags but switched to grey bags to reduce the appearance of trash receptacles in crowded public spaces. Thus, this study's findings are in opposition to the intended application by the City of Paris.

4. The authors use the phrase 'perceived trash density' to refer to the salience of trash bins, but this term mischaracterizes the physical norm. Trash density could be misinterpreted to refer to the amount of litter or trash in the physical environment. Instead, the authors should refer to this as perceived receptacle density.

Reviewer #2: This article reports the findings from three studies that assessed the impact of bin bag color on bin visibility in an online experimental setting. Using a Signal Detection Theory approach and photographs featuring bins on urban Parisian streets as stimuli, the authors tested whether changing the bin bag color from grey to red, green or blue increased bin detection rates. Studies 1 and 2 involved British participants to infer a common nationality of tourists to Paris, whereas Study 3 featured a more local Parisian sample. I found this article an interesting read, and I believe it addresses an important gap in literature in the area of environmental psychology and policy. The authors have provided a convincing example of a series of online studies that lends itself to cost-effective implications in real-life contexts. However, I have identified four main areas that require revision before the article can fulfill the criteria of publication in this journal. These revisions do not require extensive adjustments to the manuscript, and thus I have recommended minor revisions.

CRITICAL APPRAISAL

The authors haven’t discussed limitations in regard to the methodology that they have used. The studies could be seen as very leading and lacking ecological validity based on the design (i.e. telling people to look for bins and stating if they have seen them vs. not). They do note analytical techniques to try and reduce biases, but this also needs to be acknowledged in the discussion. At the moment, it implies that the findings show detectability, but that cannot be assumed to translate to everyday experiences of the Parisian streets. It would have been nice to have a field study to test it in a real life context or a laboratory study that has a less obvious leading focus, but at least the authors should acknowledge the barriers of conducting an experimental (online) study and its implications. In summary, more discussion and critical appraisal of the aforementioned points would make the article a more sound and convincing piece of scientific research.

LACK OF REFERENCES

The Introduction makes a strong case for the rationale of the study, and the authors have discussed the impacts of litter in the environment convincingly. Similarly, discussion on implications is thorough and plausible. In my view, some of the key claims, especially those in the Introduction (e.g. information about Paris’ waste management system and budget, endogenous and exogenous attention) and sections of the methods (e.g. when justifying bin bags colours) would need to be supported with references. I would recommend that the authors provide more justification to their claims by providing more references to existing literature throughout. Finally, it is surprising that nudge theory was only mentioned in the Discussion, and not in the Introduction, so an earlier reference is needed.

OMISSIONS/INCONSISTENCIES IN THE RESULTS SECTION AND DATA ANALYSIS

In general, the Method and Results sections read well and are concise, and the provided photographs that help to illustrate the experimental design. The authors have also explained the application of the methodology in the current studies very well. However, the authors’ statement of statistical power (Method section) is incoherent and superficial, as it is unclear what “a minimum effect of 10%” means. PLOS ONE guidelines state that “If a sample size calculation was performed, specify the inputs for power, effect size and alpha”.

Regarding how the authors have reported their findings, I think it important that the statistical tests are mentioned each time the results are reported (as currently ‘paired t-tests’ is only mentioned in Results for study 1), and past tense should be used when reporting findings. As PLOS ONE asks for effect sizes, these should be provided, or a justification as to why they haven’t been provided in this instance should be given. As this paper seems to be directed to decision makers, it would also be useful to qualitatively state the effect size (e.g. d = xx, a medium effect) to help emphasize the strength further.

The largest omission within the results section is the detailed given for the ANOVAs. The results from the mixed model ANOVAs are not reported in enough detail to follow the analysis, know what was done and for the reader to conclude which color bags produced the ‘larger’ changes from grey. Although the authors provide the d’ values earlier in the results section, it would still be useful to state here the directions of the interaction effects and report post-hoc analyses if relevant. Figure 4 helps with this interpretation, but clearer (and higher resolution) interaction plots could be provided to help illustrate results from the mixed model ANOVA.

The authors should also note that the Interpretation of the findings for reaction times on page 16 seem incorrect, stating reaction time was shorter for the colored condition, but then Study 2B the colored condition was slower (1056 seconds vs. 1006 seconds). Moreover, while all other studies report what the equivalent of the percentage difference would be in terms of number of new bins installed in Paris, this information is not provided for the blue bag condition (Study 2B).

Another key point is that the authors have referred to potentially new findings in the Discussion (asking participants which color they would pick). Is this already published, thus needing a citation, or is this unpublished work? This information either needs to be citeable or included as a study in the paper, as no new findings should be introduced in the Discussion.

WRITTEN EXPRESSION AND FORMATTING

The paper is mostly written in clear and consistent way. However, the authors should note that “zero-cost” is a strong statement for this paper. As the purpose of the paper wasn’t to run a cost-benefit analysis, it looks like this is based on an assumption (for example, might colored bags be charged differently, need different suppliers, harder to source?). It is also important to use neutral terms (“manpower” could be changed to ‘resources’ in the Introduction) and avoid idiomatic expressions (“down-the-road” could be changed to ‘subsequent’ in the Discussion). Similarly, “promising” (p. 9) needs to be elaborated on for clarity as in its current format it appears contradictory. In regard to formatting, use of line numbers and double spacing throughout would have made the article easier to read and review. Similarly, it is important to be consistent with formatting of paragraph transitions. Finally, the paper should be checked for typing errors and clarity – Some typos, formatting errors and other points that need elaboration are listed below:

o \\the and /word” (\\ is sometimes used in place of “)

o SD should be used instead of +/- (when reporting age of participants)

o p < :001 should be p < 0.001 (PLOS ONE guidelines)

o be consistent with reporting the statistics – 2 decimal points

o close bracket missing on page 16

o Some numbers don’t add up (p. 12 (104-204 = 309) might mean one participant did not report their gender, but needs to be stated.)

o As the studies were not ran under highly controlled laboratory conditions, they should not be referred to as laboratory studies (maybe experimental online studies)

o Paragraph 5/6 in the Introduction: Should be reworded to better reflect that bins that blend with the environment are NOT a good idea

o Were people who reported being color blind excluded from the study? There’s reference this information was collected, but did not state how the authors used it.

o Abstract: It is not clear to a reader who hasn’t read the full text whether the “replication studies” mentioned at the end are included in or separate from the “three pre-registered studies” mentioned at the beginning.

Reviewer #3: The manuscript reports an experiment based on internet data collection on a visual detection task of trash bins within a complex visual scene. The experimental design is a within-subjects in which the colour of the visual target is manipulated. The results show that when the target is coloured the sensitivity of the subject is higher than when the target is grey. The result is replicated with different colours, and it turns out that blue is the one that favours a higher detectability of the target. Furthermore, a final study shows that the effect is present regardless of whether participants are previously familiar with the target or with the complex visual scene.

The idea that colour shade can influence the detection (and identification) of visual stimuli is not particularly new, but the scope of this study is. I found the manuscript interesting and well written. The methodology is sound. In fact, the visual detection paradigm is applied to a realistic context, the images are cityscapes of Paris, and the target are the city's trash bins. I agree with the authors on the relevance of the study which demonstrates how changing the colour of the bin bags from the current grey to blue can increase bin detection and potentially improve urban decency.

However, I feel that some of the claims (particularly in the discussion) are speculative. Although the dependent variable is the detection accuracy for individual bins in cityscape images, the authors refer several times in the text to the perceived density of bins. Since the latter is never measured directly or indirectly, I think the authors should revisit the claim that changing the colour of the bag produces an increase in the perceived density of the bins. If the authors assume a linear relationship between detection accuracy and the perceived density of the bins in the visual scene, they should at least substantiate this relationship with other data or with support from previous literature, since they only have one bin per image in this experiment. Perceived density may in fact depend not only on detectability, but also on salience and other contextual factors. The same applies to the conversion between the increase in detection accuracy and the absolute number of bins in the city and consequently also to the equivalent economic savings. In fact, the data analysed refer to two-dimensional images with exposure times of 700 milliseconds. as the authors were interested in measuring the bottom-up effect related to colour. What would happen without such high time pressure? In a real-life situation there is usually sufficient time for serial scanning of scene elements and this could greatly reduce the difference in detection rate between coloured and grey bins. Therefore, quantifying economically the savings in terms of number of bins based on the data of this experiment which is limited to a very specific condition could be misleading. It is not known how much the detection rate would increase under real conditions, nor the perceived density of bins (in the virtual or real environment). Although reporting how many euros would be saved is suggestive, the exact amount remains speculative. I would recommend the article for publication only after the authors have more cautiously revised the sentences that are not directly supported by the experimental data.

Have you checked that the assumptions for ANOVAs were met? You should report this in a sentence in the analysis section.

You should also report if any correction has been applied and you should also better specify the t.test (is it always equal variance two-tailed t-test?). When doing multiple test on the same dataset you should perform a correction for multiple comparisons (for example when testing green vs red vs blue in pairs).

Sometimes it is not clear which are the factors and the levels of the factor in the ANOVAs:

In the ANOVA for COLOUR x STUDY, how many COLOUR factor levels are there? one would expect a factor with 3 levels (red, green, and blue) and thereafter a series of pair-wise post-hoc with correction for multiple comparisons, but you seem to have done 3 separate ANOVAs, is there a particular reason?

In experiment 3 which are the factors (and the levels for each factor) for the mixed ANOVA?

Minor

I think there are some inconsistencies in the notation (e.g. sometimes d'0, .d0, d0, etc.).

I think that the dot and not the colon should go before the decimal part of the p value.

When the SDT indices are described, there are slashes instead of quotation marks.

The d'0 represents the sensitivity of the participant to the stimulus. Indeed, for the same signal-to-noise ratio d'0 can vary from participant to participant. Although d' tends to increase as the signal-to-noise ratio increases (the signal and noise probability curves get further apart) I would not expressly say that d' represents the signal-to-noise ratio, but rather the sensitivity of the measuring instrument or the person making the detection.

It is true that “participants did not have enough time to fully scan the scene in a top-down fashion” but at the same time the task was based on a voluntary active search involving also endogenous attention. From the introduction it seems that the task focuses on endogenous attention, but endogenous attention takes about 300 ms to be deployed and can be sustained at will whereas exogenous attention takes only about 100 ms to be deployed and it is transient (for reviews see Carrasco 2015 or Giordano 2009). Given that average reaction times are above one second this would suggest a deliberate serial scanning of the visual scene, but the fact that there is an increase in performance and also a reduction in reaction times in the 'colour' condition suggests a greater involvement of exogenous attention. It remains to be seen whether prolonging the exposure time further will cause the advantage to lapse (how is the performance in catch trials?). In real-life situations we rarely have less than a second to identify the bin. I think the part about attention and reaction time is lacking in the discussion.

The euro symbol is not always written correctly and sometimes the thousand separator is missing in numbers

6. PLOS authors have the option to publish the peer review history of their article (what does this mean?). If published, this will include your full peer review and any attached files.

Reviewer #1: No

Reviewer #2: No

Reviewer #3: **Yes: **Giulio Contemori

---

## [Author Response · Author response to Decision Letter 0]

13 Jan 2022

Dear editor,

Thank you for giving us the opportunity to submit a revised draft of our manuscript. We appreciate the time and effort that you and the reviewers have dedicated to providing your feedback on our manuscript. We have incorporated the reviewers’ suggestions and highlighted the changes in red within the manuscript.

You will find below a point-by-point response to the reviewers’ comments.

Reviewer #1: While the topic of the study was exciting, the results do not warrant publication in their current form. Listed below are my key concerns, and unfortunately, I do not believe that these issues can be addressed without additional data.

1. The critical missing piece is data showing the connection between bin density and litter. The authors make a fundamental assumption that higher bin density will result in lower littering rates, but this is not substantiated. In fact, from the social psychological research on normative influence, there is reason to predict that increased bin density could spark MORE litter in environments that have a high degree of existing litter (see research by Cialdini on the focus theory of normative conduct).

The focus theory of normative conduct predicts that a higher number of bins or more visible trash cans, coupled with high levels of litter, might induce increased littering. However, empirical studies suggest that making trash containers more attractive or noticeable can work as an effective alternative to adding bins (Arnold, 2015; Geller et al., 1980; O’Neill, Blanck & Joyner, 1980). More visible bins are also used more, regardless of their availability, which suggests that an increase in saliency can be an effective policy to decrease littering down the line. Our goal was to provide an objective way of measuring which colours are more likely to increase salience.

We altered the sentence in the introduction to include Cialdini’s reference 5 (line 51): 

“While some theories suggest that an increase in the number of bins might increase litter in the presence of litter in the environment (11), most empirical studies that have focused on understanding and preventing the widespread prevalence of littering as early as in the 1970s have confirmed that sites with more receptacles tend to have lower littering rates (12–15).”

2. More clarity is needed about the theoretical foundations of this research. The authors briefly mention the Signal Detection Theory, and later 'Nudge Theory', yet the results are not integrated into these two foundations. Specifically, Study 3 revealed the strongest increases in perceptions of receptacles as a result of switching to blue bags, compared to green and red bags. However, the authors do not address that this result is contradictory to the signal detection theory's claims for increased attention to the color red.

To the best of our knowledge, signal detection theory does not claim that the colour red necessarily increases attention. Exogenous attention is referred to as bottom-up salience and is typically triggered by contrast differences between objects and their surroundings, e.g., if the colour of an object is particularly salient compared to the background against which it stands. Therefore, a blue object among various red objects will stand out more than a red object. 

We clarified this in the introduction (line 94): 

“Exogenous attention is also referred to as bottom-up salience and typically emerges from contrast differences between objects and their surroundings; for example, if the colour of an object is particularly salient compared to the background against which it stands (e.g., a blue object against a red background will be salient).”

In addition, while Nudge Theory is cited, there is a body of research in social psychology around norms that is highlight relevant and should be integrated into the paper.

Nudges offer ways to alter the surroundings of individuals and directing them towards the desired outcome using cognitive levers such as priming (lemon scented trains to decrease litter (DeLange et al., 2012) or social norms (displaying eye images to decrease litter (Bateson et al., 2011)). We agree that social psychology also provides a relevant body of research which is now cited more extensively. However, we would like to emphasize that our method focuses on a measure of exogenous attention, which is likely less susceptible to social norm effects. 

However, we acknowledge the Reviewer’s comment and add the following sentence to further clarify things (line 100): 

“Interventions in litter control, such as ex-ante campaigns reminding individuals of littering costs or highlighting the social norm or high fines that punish uncivil behaviour ex-post, might increase endogenous attention and trigger top-down visual search to find the closest bin (23,24). Interventions aiming at increasing the salience of bins relative to the surroundings will, on the other hand, trigger automatic attentional capture to the bin. Such an intervention could take the form of a simple change of trash bag colour from grey to more salient colours, leading to an increase in the perceived number of bins.”

3. One of the implications of this research is support for supplementing the cost associated with increased trash bins compared with increased salience of existing bins. However, the authors mention that the city previously utilized green bags but switched to grey bags to reduce the appearance of trash receptacles in crowded public spaces. Thus, this study's findings are in opposition to the intended application by the City of Paris.

In the touristic city of Paris, there is a trade-off between the aesthetics of the bins and litter control, and the color of bins should be changed most efficiently to maximize the visibility for the aesthetic cost. There is an ongoing debate between officials who vouch to increase the number of bins or to make them more visible and those who vouch for making bins more invisible for aesthetic reasons. Even though the city of Paris has previously switched the colour of bin bags to grey, there was no evaluation on whether this had produced more or less litter, and as the deliberation is ongoing, the city was interested in having more evidence to figure out whether this had had any effect on the visibility of bins. 

4. The authors use the phrase 'perceived trash density' to refer to the salience of trash bins, but this term mischaracterizes the physical norm. Trash density could be misinterpreted to refer to the amount of litter or trash in the physical environment. Instead, the authors should refer to this as perceived receptacle density.

We thank the reviewer for pointing out this error. The occurrence of “perceived trash density” line 110 was changed to “perceived bin density” to match the rest of the sentences in the paper.

Reviewer #2:

This article reports the findings from three studies that assessed the impact of bin bag color on bin visibility in an online experimental setting. Using a Signal Detection Theory approach and photographs featuring bins on urban Parisian streets as stimuli, the authors tested whether changing the bin bag color from grey to red, green or blue increased bin detection rates. Studies 1 and 2 involved British participants to infer a common nationality of tourists to Paris, whereas Study 3 featured a more local Parisian sample. I found this article an interesting read, and I believe it addresses an important gap in literature in the area of environmental psychology and policy. The authors have provided a convincing example of a series of online studies that lends itself to cost-effective implications in real-life contexts. However, I have identified four main areas that require revision before the article can fulfill the criteria of publication in this journal. These revisions do not require extensive adjustments to the manuscript, and thus I have recommended minor revisions.

CRITICAL APPRAISAL

The authors haven’t discussed limitations in regard to the methodology that they have used. The studies could be seen as very leading and lacking ecological validity based on the design (i.e. telling people to look for bins and stating if they have seen them vs. not). They do note analytical techniques to try and reduce biases, but this also needs to be acknowledged in the discussion. At the moment, it implies that the findings show detectability, but that cannot be assumed to translate to everyday experiences of the Parisian streets. It would have been nice to have a field study to test it in a real life context or a laboratory study that has a less obvious leading focus, but at least the authors should acknowledge the barriers of conducting an experimental (online) study and its implications. In summary, more discussion and critical appraisal of the aforementioned points would make the article a more sound and convincing piece of scientific research.

We thank the reviewer for this comment. We would like to stress that the aim of the study was to tap bottom-up attention, or how much is a bin seen when the passerby is precisely not looking for one. A very short display time helps us tap this process as the person does not have time to direct their eyes more than 2 or 3 times. In the discussion, we acknowledge that we cannot infer much about real-life conditions where people are actively looking for a bin and leverage top down attention. Additional field experiments are needed to test the combined effects of bottom-up and top-down attention. 

This was clarified in the discussion (line 425):

"Our results also provide grounds for future work investigating whether an increased visibility of bins has a subsequent impact on littering behaviour but a number of limitations should be acknowledged. First, our set of experimental online studies may lack ecological validity and looking for a bin in real settings may rely on different mechanisms. Specifically, the short display time in our experiment mimics the impact of increased visibility when people are not actively looking for a bin, which differs from what happens in real-life. Moreover, our studies do not allow us to state whether changing bag colour would have an effect on real-life littering. However, prior research provides suggestive evidence that making bins more attractive or noticeable affects behavior (17–19).”

LACK OF REFERENCES

The Introduction makes a strong case for the rationale of the study, and the authors have discussed the impacts of litter in the environment convincingly. Similarly, discussion on implications is thorough and plausible. In my view, some of the key claims, especially those in the Introduction (e.g. information about Paris’ waste management system and budget, endogenous and exogenous attention) and sections of the methods (e.g. when justifying bin bags colours) would need to be supported with references. I would recommend that the authors provide more justification to their claims by providing more references to existing literature throughout. 

The following references have been added in support of the claims made by the authors. 

“La droite donne un an à Hidalgo pour améliorer la propreté de Paris [Internet]. Les Echos. 2018 [cité 1 sept 2021]. Disponible sur: https://www.lesechos.fr/2018/02/la-droite-donne-un-an-a-hidalgo-pour-ameliorer-la-proprete-de-paris-983415”

« Berger A, Henik A, Rafal R. Competition Between Endogenous and Exogenous Orienting of Visual Attention. Journal of Experimental Psychology: General. 2005;134(2):207‑21. »

We cannot provide more references about the choice of colours since it came from exchanges during meetings with the local authorities with whom we worked on this project.

Finally, it is surprising that nudge theory was only mentioned in the Discussion, and not in the Introduction, so an earlier reference is needed.

Indeed, it would be useful to mention nudge in the introduction to contextualise the study. The following sentence was added in the introduction (line 74):

“In line with research on nudges, such simple changes to the physical surroundings can be effective in significantly changing behaviour (20)”

OMISSIONS/INCONSISTENCIES IN THE RESULTS SECTION AND DATA ANALYSIS

In general, the Method and Results sections read well and are concise, and the provided photographs that help to illustrate the experimental design. The authors have also explained the application of the methodology in the current studies very well. However, the authors’ statement of statistical power (Method section) is incoherent and superficial, as it is unclear what “a minimum effect of 10%” means. PLOS ONE guidelines state that “If a sample size calculation was performed, specify the inputs for power, effect size and alpha”. 

We acknowledge that the power calculation reporting was opaque. The following paragraph was added to clarify (line 203): 

“...which is a large enough sample to detect a small effect size of dz=0.15 (α=5%, 1-β=80%) following a power analysis with G*Power (version 3.1.9.3) for a two-tailed t-test for dependent means, and using outcome variances computed in the pilot study over 100 participants.”

Regarding how the authors have reported their findings, I think it is important that the statistical tests are mentioned each time the results are reported (as currently ‘paired t-tests’ is only mentioned in Results for study 1).

We thank the reviewer for drawing our attention to this oversight, which is now corrected in the Result sections of Studies 2.A, 2.B and 4.

“and past tense should be used when reporting findings.”

The present tense was changed to the past tense in all result sections. 

“As PLOS ONE asks for effect sizes, these should be provided, or a justification as to why they haven’t been provided in this instance should be given. As this paper seems to be directed to decision makers, it would also be useful to qualitatively state the effect size (e.g. d = xx, a medium effect) to help emphasize the strength further.”

We added all effect sizes to the results sections. 

“...(cohen’s d=0.88, a large effect)” (line 252)

“...(cohen’s d=0.45, a medium effect)” (line 255)

“...(cohen’s d=1, large effect size)” (line 299)

“This is a very large effect size with Cohen’s d for repeated measures equaling 1.55.” (line 307)

“(cohen’s d for repeated measured=1.5, large effect size)” (line 375)

The largest omission within the results section is the detailed given for the ANOVAs. The results from the mixed model ANOVAs are not reported in enough detail to follow the analysis, know what was done and for the reader to conclude which color bags produced the ‘larger’ changes from grey. Although the authors provide the d’ values earlier in the results section, it would still be useful to state here the directions of the interaction effects and report post-hoc analyses if relevant. Figure 4 helps with this interpretation, but clearer (and higher resolution) interaction plots could be provided to help illustrate results from the mixed model ANOVA.

We acknowledge the results concerning the mixed ANOVA were unclear. To help illustrate the results, we made adjustments to the results section and replaced the figure with the following interaction graph, as suggested by the reviewer.

Fig 4. Interaction graph showing the change in detectability when a bin is coloured, in studies 1 and 2. Mean and 95% confidence interval. 

The authors should also note that the Interpretation of the findings for reaction times on page 16 seem incorrect, stating reaction time was shorter for the colored condition, but then Study 2B the colored condition was slower (1056 seconds vs. 1006 seconds). 

We thank the reviewer for pointing out this mistake. We had inverted the previously erroneous reaction times and have now corrected this in the manuscript.

Moreover, while all other studies report what the equivalent of the percentage difference would be in terms of number of new bins installed in Paris, this information is not provided for the blue bag condition (Study 2B).

We thank the reviewer for pointing out this unintentional omission. We have added the following (line 311): 

This translates into an increase of 46% of trash can visibility, similar to the addition of 13,600 new bins.” 

Another key point is that the authors have referred to potentially new findings in the Discussion (asking participants which color they would pick). Is this already published, thus needing a citation, or is this unpublished work? This information either needs to be citeable or included as a study in the paper, as no new findings should be introduced in the Discussion. 

We added a section in the paper to report these findings as a separate study.

“Study 3: Which colour would you have picked?

The superiority of the colour blue to the colour red in the bin detection tasks did not follow our initial intuition, that the colour red would be most visible. We thus ask a random sample of participants which colour they would choose if they were policy makers. This could be an important point since policy makers often base important choices on intuition, which might be highly inefficient. 

Participants and trial

160 British nationals were recruited on the platform Prolific Academic and received a compensation of 62 euros per hour. Participants were only asked one question: “Imagine you are in charge of making sure that bins in your city are highly visible. Which colour would you pick for the bins? Green, red or blue?''. The choices appeared in a random order to participants in order to avoid any biases in answers. 

Results and Discussion

The most common answer was red (50%), followed by blue (32%) and green (18%). This was consistent with our intuition and some studies on visual attention that red is believed to be the most salient (1). Assuming that policy makers’ intuition is comparable, this might lead Parisian policy makers to choose a sub-optimal solution when choosing the colour that best increases the saliency of a bin in the city.”

Additionally, we changed the paragraph in the discussion, to: 

“Indeed, if a policy maker wants to increase the perceived density of bins, it would be inefficient to go straight to a costly field trial comparing various bin colours. And as our additional data show, it would be even more inefficient to simply rely on intuitions. When we asked a sample of different participants what colour they believe would be the most efficient at increasing bin visibility, the majority answered “red” while our studies show that blue is in fact the colour that best improves detectability. Determining which colour is most likely to be salient in a particular city in an experimental setting first can therefore be a useful evidence-based first step before elaborating a field strategy.” 

WRITTEN EXPRESSION AND FORMATTING

The paper is mostly written in clear and consistent way. However, the authors should note that “zero-cost” is a strong statement for this paper. As the purpose of the paper wasn’t to run a cost-benefit analysis, it looks like this is based on an assumption (for example, might colored bags be charged differently, need different suppliers, harder to source?). 

We acknowledge that the term “zero-cost” might not be generalizable to all contexts. However, in the context of this study, our exchanges with the local authorities confirmed that the red, green and blue bags were provided by the same suppliers and cost the same as grey bags, so if the authorities changed their bags from grey to red, the cost would truly be zero. 

“Footnote 1. In the context of this study, our exchanges with the Parisian local authorities confirmed that the red, green and blue bags were provided by the same suppliers and cost the same as grey bags, so if the authorities changed their bags from grey to red, the cost would be zero. This, however, does not necessarily hold in all contexts where the production of a new coloured bag might incur a relatively low, but non-zero cost.”

It is also important to use neutral terms (“manpower” could be changed to ‘resources’ in the Introduction) and avoid idiomatic expressions (“down-the-road” could be changed to ‘subsequent’ in the Discussion). 

We thank the reviewer for these suggestions. The two substitutions have been applied as suggested.

Similarly, “promising” (p. 9) needs to be elaborated on for clarity as in its current format it appears contradictory. 

We agree with the reviewer that the word might create confusion. It has been substituted with “relevant”.

In regard to formatting, use of line numbers and double spacing throughout would have made the article easier to read and review. 

Similarly, it is important to be consistent with formatting of paragraph transitions. 

The formatting has been adjusted accordingly and line numbers + double spacing have been applied.

Finally, the paper should be checked for typing errors and clarity – Some typos, formatting errors and other points that need elaboration are listed below: 

\\the and /word” (\\ is sometimes used in place of “).SD should be used instead of +/- (when reporting age of participants).p <:001 should be p < 0.001 (PLOS ONE guidelines).be consistent with reporting the statistics – 2 decimal points.close bracket missing on page 16. Some numbers don’t add up (p. 12 (104-204 = 309) might mean one participant did not report their gender, but needs to be stated.). As the studies were not ran under highly controlled laboratory conditions, they should not be referred to as laboratory studies (maybe experimental online studies)

All of the typos have been corrected and all suggestions were taken into account and altered in the text. 

Paragraph 5/6 in the Introduction: Should be reworded to better reflect that bins that blend with the environment are NOT a good idea.

Indeed, the paragraph should be rephrased in order to better reflect the trade-off. The last two sentences were changed to (line 78): 

“This solution appears particularly relevant in cities, where policy makers strive to keep streets charming and deliberately favour designs that make bins blend with the surrounding urban environment. In Paris, for instance, public trash bins are made of a light grey metal and fitted with discreet grey bags, which means that they are essentially designed to be invisible (Fig 1).”

Were people who reported being color blind excluded from the study? There’s reference this information was collected, but did not state how the authors used it.

Color blind participants were not excluded from the analysis, since we believe this improves the generalizability of the results to real-life situations. The results were used to make sure that the rate of colour-blind participants was not higher than the one found in the population. 

We clarified this point in the paper in all “Participants” sections: 

“The share of color blind participants was similar to that observed in the United Kingdom.”

Abstract: It is not clear to a reader who hasn’t read the full text whether the “replication studies” mentioned at the end are included in or separate from the “three pre-registered studies” mentioned at the beginning.

The abstract was changed to clarify this point: 

“In an effort to inform interventions targeting littering behaviour, we estimate how much a change in trash-bag colour increases trash can visibility in Paris. To that end, we applied standard Signal Detection techniques to test how much changing trash-bag colour from grey to red affects subjects' detection rates. In three pre-registered studies (total N = 922), we found that changing trash bag colour considerably increases the perceived number of bins: a change from grey to red translates into a 28% increase in the perceived number of bins (Study 1). When considering changes in bottom up attention, this means that a zero-cost change of trash-bag colour from grey to red could be equivalent to installing up to 8,400 additional bins in the city of Paris, in terms of perceived density. Two replication studies investigating other colour changes showed that changing the colour from grey to blue further increases visibility, with blue exhibiting the highest increase in visibility in a sample living in the Paris area.”

Reviewer #3: The manuscript reports an experiment based on internet data collection on a visual detection task of trash bins within a complex visual scene. The experimental design is a within-subjects in which the colour of the visual target is manipulated. The results show that when the target is coloured the sensitivity of the subject is higher than when the target is grey. The result is replicated with different colours, and it turns out that blue is the one that favours a higher detectability of the target. Furthermore, a final study shows that the effect is present regardless of whether participants are previously familiar with the target or with the complex visual scene.

The idea that colour shade can influence the detection (and identification) of visual stimuli is not particularly new, but the scope of this study is. I found the manuscript interesting and well written. The methodology is sound. In fact, the visual detection paradigm is applied to a realistic context, the images are cityscapes of Paris, and the target are the city's trash bins. I agree with the authors on the relevance of the study which demonstrates how changing the colour of the bin bags from the current grey to blue can increase bin detection and potentially improve urban decency.

However, I feel that some of the claims (particularly in the discussion) are speculative. Although the dependent variable is the detection accuracy for individual bins in cityscape images, the authors refer several times in the text to the perceived density of bins. Since the latter is never measured directly or indirectly, I think the authors should revisit the claim that changing the colour of the bag produces an increase in the perceived density of the bins. If the authors assume a linear relationship between detection accuracy and the perceived density of the bins in the visual scene, they should at least substantiate this relationship with other data or with support from previous literature, since they only have one bin per image in this experiment. Perceived density may in fact depend not only on detectability, but also on salience and other contextual factors. The same applies to the conversion between the increase in detection accuracy and the absolute number of bins in the city and consequently also to the equivalent economic savings. 

In fact, the data analysed refer to two-dimensional images with exposure times of 700 milliseconds. as the authors were interested in measuring the bottom-up effect related to colour. What would happen without such high time pressure? In a real-life situation there is usually sufficient time for serial scanning of scene elements and this could greatly reduce the difference in detection rate between coloured and grey bins. Therefore, quantifying economically the savings in terms of number of bins based on the data of this experiment which is limited to a very specific condition could be misleading. It is not known how much the detection rate would increase under real conditions, nor the perceived density of bins (in the virtual or real environment). Although reporting how many euros would be saved is suggestive, the exact amount remains speculative. I would recommend the article for publication only after the authors have more cautiously revised the sentences that are not directly supported by the experimental data.

We agree with the author that these claims are speculative and they are only for making concrete comparisons to policy makers, but are based on assumptions. The tone of the discussion has thus been changed to imply this, and the approximated economic gains have been removed throughout the study (line 405): 

“The results of this study suggest that a mere change in the colour of bins can result in a cost-effective intervention that would increase the visibility of bins. This implies that some previously invisible bins would become visible as a result of the intervention, thereby increasing the perceived density of bins in an urban setting. Assuming a linear relationship, and for illustration, we speculate that a change of bag colour from grey to red, green or blue in the city of Paris could be equivalent to installing up to 8,370, 9,537 or 13,600 additional bins, respectively.”

Additionally, in the abstract, the phrase “considerably increases the perceived density of bins” was changed to “considerably increases the perception of bins”

Have you checked that the assumptions for ANOVAs were met? You should report this in a sentence in the analysis section.

We checked that the ANOVA assumptions of normality, independence of samples, and sphericity were met. However, we did not have homogeneity of variances at all treatment level (Levene’s test, p < .05). We therefore performed a robust ANOVA using the R package WRS2. The results section was fixed accordingly (line 319): 

“The ANOVA assumptions of normality, independence of samples and sphericity were met but variances were heterogeneous. We therefore ran a mixed-model robust ANOVA using the R package WRS2, with 3-level factor STUDY (green, red and blue) and 2-level factor COLOUR (grey, colour).” 

You should also report if any correction has been applied and you should also better specify the t.test (is it always equal variance two-tailed t-test?). When doing multiple test on the same dataset you should perform a correction for multiple comparisons (for example when testing green vs red vs blue in pairs).

All instances of t.test were substituted for "paired equal variance two-tailed t-test for repeated measures” to better specify the t-test conducted. 

Sometimes it is not clear which are the factors and the levels of the factor in the ANOVAs: In the ANOVA for COLOUR x STUDY, how many COLOUR factor levels are there? One would expect a factor with 3 levels (red, green, and blue) and thereafter a series of pair-wise post-hoc with correction for multiple comparisons, but you seem to have done 3 separate ANOVAs, is there a particular reason?

The reason why 3 separate ANOVAs with 2 factors were conducted after the ANOVA with 3 factors was to be able to capture a change in the "effect size" or the difference between grey and color between two factors. A pairwise post hoc test would allow a comparison of the three colors at each condition (color and grey) but not of the change from grey to color at each factor level.

We hope to have clarified this in the paper (line 317): 

" We performed a mixed-model ANOVA to compare the results obtained in Studies 1, 2.A and 2.B, using the change from grey to colour as a within subject factor and study number as the between subjects factor. The ANOVA assumptions of normality, independence of samples and sphericity were met but variances were heterogeneous. We therefore ran a mixed-model robust ANOVA using the R package WRS2, with 3-factor levels STUDY (green, red and blue) and 2-factor levels COLOUR (grey, colour). We found a significant interaction F(912)=22.672 p=0.002. We then did three separate mixed ANOVAs to compare the effect size of COLOUR, this time with a 2-factor levels STUDY. An interaction graph (Fig 4) showed the change from grey to colour across all three studies. We found significant COLOUR x STUDY interactions when comparing red (Study 1) to blue (Study 2.B) F (1, 615) =35.92 p<0.001 and when comparing green (Study 2.A) to blue (Study 2.B) F (1, 610) =32.01 p<0.001; but no interaction when comparing red (Study1) to green (Study 2.A), F (1, 611) =0.46 p = 0.45. "

In experiment 3 which are the factors (and the levels for each factor) for the mixed ANOVA? 

We thank the reviewer for his observation. The following was added to explain further (line 391): 

"A mixed model ANOVA testing 2-level COLOUR factor (grey vs colour) x 3-level STUDY factor (study 2.B vs study 4)." 

Minor

I think there are some inconsistencies in the notation (e.g. sometimes d'0, .d0, d0, etc.)

All annotations were changed to d^'.

I think that the dot and not the colon should go before the decimal part of the p value.

We thank the reviewer for pointing this out. The colon was changed into a dot.

When the SDT indices are described, there are slashes instead of quotation marks.

We corrected this unintentional typo. 

The d'0 represents the sensitivity of the participant to the stimulus. Indeed, for the same signal-to-noise ratio d'0 can vary from participant to participant. Although d' tends to increase as the signal-to-noise ratio increases (the signal and noise probability curves get further apart) I would not expressly say that d' represents the signal-to-noise ratio, but rather the sensitivity of the measuring instrument or the person making the detection.

We agree that this needed to be reworded (line 224): 

“A discriminability of 0 means that the signal is not distinguished from noise, while higher d0 values represent a higher sensitivity (i.e. a situation where it is easier for people to discriminate the bin in the scene)”. 

It is true that “participants did not have enough time to fully scan the scene in a top-down fashion” but at the same time the task was based on a voluntary active search involving also endogenous attention. From the introduction it seems that the task focuses on endogenous attention, but endogenous attention takes about 300 ms to be deployed and can be sustained at will whereas exogenous attention takes only about 100 ms to be deployed and it is transient (for reviews see Carrasco 2015 or Giordano 2009). Given that average reaction times are above one second this would suggest a deliberate serial scanning of the visual scene, but the fact that there is an increase in performance and also a reduction in reaction times in the 'colour' condition suggests a greater involvement of exogenous attention. 

It remains to be seen whether prolonging the exposure time further will cause the advantage to lapse (how is the performance in catch trials?). In real-life situations we rarely have less than a second to identify the bin. I think the part about attention and reaction time is lacking in the discussion.

We agree with the reviewer that our data suggest a greater involvement of exogenous attention, which is consistent with our interpretation and task design. Our goal was to focus on automatic and rapid attentional processes so we made sure that the stimulus presentation time only enabled 2 to 3 fixations (700 ms). Given the complexity of the visual scene, participants did not have enough time to fully scan the scene. Reaction times were indeed above one second on average, but reaction times capture several processes beyond attention, including the selection of motor goals and motor planning. We clarified this in the Results and Discussion section of Study 1 (line 258):

“However, it is worth pointing out that the reaction times capture several processes beyond attention, including the selection of motor goals and motor planning, which explains why they are on average above one second even though attentional capture happens much faster.” 

We also would like to stress that the aim of the study was to tap bottom-up attention, or how much is a bin seen when the passerby is precisely not looking for one. In the discussion, we acknowledge that we cannot infer much about real-life conditions where people are actively looking for a bin and leverage top down attention. Additional field experiments are needed to test the combined effects of bottom-up and top-down attention. 

This was clarified in the discussion:

"Our results also provide grounds for future work investigating whether an increased visibility of bins has a subsequent impact on littering behaviour but a number of limitations should be acknowledged. First, our set of experimental online studies may lack ecological validity and looking for a bin in real settings may rely on different mechanisms. Specifically, the short display time in our experiment mimics the impact of increased visibility when people are not actively looking for a bin, which differs from what happens in real-life. Moreover, our studies do not allow us to state whether changing bag colour would have an effect on real-life littering. However, prior research provides suggestive evidence that making bins more attractive or noticeable affects behavior (17–19).”

The euro symbol is not always written correctly and sometimes the thousand separator is missing in numbers

We thank the reviewer for pointing that out. With the removal of the figures related to financial gains, this has been adjusted in the manuscript.

---

## [Decision Letter · Decision Letter 1]

11 Apr 2022

PONE-D-21-08195R1A zero-cost attention-based approach to promote cleaner streets: a Signal Detection Theory approach in Parisian streetsPLOS ONE

Dear Dr. Abdel Sater,

thank you for submitting your revised manuscript to PLOS ONE. After careful consideration by four experts in the field, we feel that despite their diverging opinions, it has merit but does not fully meet PLOS ONE’s publication criteria as it currently stands. Therefore, we invite you to submit a revised version of the manuscript that addresses the points raised during the review process.

Please make sure in your revised version, to address Reviewers' comments in the point-by point manner, paying particular attention to the statistical and technical aspects of your work as mentioned in Reviewers' reports, and especially, the concerns raised by Reviewers 1 (including their previous report) and 4 below.

Please submit your revised manuscript within six months fom this date as afterwards, any revision has to be considered as a new submission. If you will need more time than this to complete your revisions, please reply to this message or contact the journal office at plosone@plos.org. Please include the following items when submitting your revised manuscript:A rebuttal letter that responds to each point raised by the academic editor and reviewer(s). You should upload this letter as a separate file labeled 'Response to Reviewers'.A marked-up copy of your manuscript that highlights changes made to the original version. You should upload this as a separate file labeled 'Revised Manuscript with Track Changes'.An unmarked version of your revised paper without tracked changes. You should upload this as a separate file labeled 'Manuscript'.

We look forward to receiving your revised manuscript. Thank you for choosing PLOS ONE for reporting your research.

Kind regards,

Sasha

Alexander N. 'Sasha' Sokolov, Ph.D.

Academic Editor

PLOS ONE

Reviewers' comments:

Reviewer's Responses to Questions

**Comments to the Author**

1. If the authors have adequately addressed your comments raised in a previous round of review and you feel that this manuscript is now acceptable for publication, you may indicate that here to bypass the “Comments to the Author” section, enter your conflict of interest statement in the “Confidential to Editor” section, and submit your "Accept" recommendation.

Reviewer #1: (No Response)

Reviewer #2: (No Response)

Reviewer #3: (No Response)

Reviewer #4: (No Response)

2. Is the manuscript technically sound, and do the data support the conclusions?

Reviewer #1: (No Response)

Reviewer #2: Partly

Reviewer #3: Partly

Reviewer #4: Partly

3. Has the statistical analysis been performed appropriately and rigorously? 

Reviewer #1: (No Response)

Reviewer #2: No

Reviewer #3: Yes

Reviewer #4: I Don't Know

4. Have the authors made all data underlying the findings in their manuscript fully available?

Reviewer #1: (No Response)

Reviewer #2: Yes

Reviewer #3: Yes

Reviewer #4: No

5. Is the manuscript presented in an intelligible fashion and written in standard English?

Reviewer #1: (No Response)

Reviewer #2: Yes

Reviewer #3: Yes

Reviewer #4: Yes

6. Review Comments to the Author

Reviewer #1: (No Response)

Reviewer #2: The authors have put considerable effort into revising their manuscript “A zero-cost attention-based approach to promote cleaner streets”, and as a result, the paper has improved. As mentioned in my first review, this paper provides a novel contribution to literature, and the rationale and implications of the conducted research are now discussed in a more convincing manner in the revised version. More revisions are needed (specifically relating to the stats and some areas needing further elaboration), therefore I have recommended minor revisions.

There are some inconsistencies in the Results reporting which I would like to draw the authors’ attention to. First, the authors need to clarify how they obtained the reported Cohen’s d effect sizes. The reported effect sizes do not match my own calculation of Cohen’s d when using the provided means and SD and exceed the feasible range (and thus the reportings of large effect sizes are also not in line with the data). Second, more information is still needed for the ANOVA results. Specifically, the reporting of factor levels doesn’t seem consistent: row 326 ->: why is STUDY now a 2-level factor? row 393 ->: “3-level STUDY factor (study 394 2.B vs study 4)”). All effects need to be fully reported: the main effect of STUDY and COLOR as well as the currently reported interaction. The justification of the follow-up analysis needs to also be given: by running separate mixed ANOVAs, familywise error is not being controlled for. Either consider running post-hoc analysis that is appropriate for these data, or provide a rationale for the chosen analysis. Finally, the statistics need to be fully reported (e.g. following APA guidelines, such as providing the effect size (partial eta squared)).

In terms of the reviewed literature and theory, the authors have now mentioned nudges in the Introduction, which is a necessary addition. However, please provide more elaboration around nudging and related literature. The paragraph starting on line 72 sounds crucial for this paper, but more elaboration is needed (a more appropriate level of depth can be found in the discussion, but this also needs to come earlier to set the scene of the paper).

Some areas of the text also need more clarification. For example, how many studies are being reported and the role of each one is not clear throughout the paper (e.g. in the abstract and end of the introduction). From my interpretation, there are five studies: three that tests detectability of bins using different colored bags on a British (tourist) population, another to test on a different population (residents) and one perception study. This needs to be made clearer throughout and would recommend avoiding the term “replicability studies” as I believe the studies all have their own unique contribution to promote and thus is more than just a “replicability study”.

Finally, there are a few minor omissions and typing errors that need to be resolved:

- Avoid non-scientific writing (e.g. lines 269-27, 273-274, 339) – decisions should be based on evidence, not intuition or what is thought to be “interesting”

- Unsupported claims – the addition of the notes on color blindness needs to be supported with evidence (e.g. % of the sample that were colorblind and a reference to what this figure is being compared to).

- row 132 ->: It is not clear how detectability was determined here, as the authors don’t explain what the pilot study participants actually did and this information comes later in the paper.

- Ensure past tense is used for reporting the studies (e.g. Study 1)

- row 147: “We also collected INFORMATION ON socio-demographic variables”

- row 269 ->: It would be useful if the authors provided further justification for testing additional colors, as this was provided in their response to the reviewers but not in the revised paper.

- Sentence starting on row 217 should be revised for clarity

- sometimes discriminability is referred to as d´ and sometimes d0 – be consistent

- row 289: could be clearer what this means

- row 296: is missing “308 for Study 2.B”

- check numbers – sample size varies between 305 and 300 for Study 2.A.

- Check consistency of incentives (between the studies they vary in currency ($ vs. €) and in value (7.50-62).

- Row 339: explain what you mean by a “random sample”

- Row 345: please also summarize the demographic profile of this sample

- Row 388 – incomplete bracket

- rows 451 – 455: The same sentence is written twice.

Reviewer #3: The manuscript is significantly improved in clarity and completeness of detail. I thank the authors for taking into consideration my previous comments, the changes made adequately respond to the majority of them. Nevertheless, I would like further clarification on one point. From my previous review:

“Although the dependent variable is the detection accuracy for individual bins in cityscape images, the authors refer several times in the text to the perceived density of bins. Since the latter is never measured directly or indirectly, I think the authors should revisit the claim that changing the colour of the bag produces an increase in the perceived density of the bins. If the authors assume a linear relationship between detection accuracy and the perceived density of the bins in the visual scene, they should at least substantiate this relationship with other data or with support from previous literature, since they only have one bin per image in this experiment. Perceived density may in fact depend not only on detectability, but also on salience and other contextual factors. The same applies to the conversion between the increase in detection accuracy and the absolute number of bins in the city and consequently also to the equivalent economic savings.”

As an answer to this concern in the body of the text you now report:

“Assuming a linear relationship, and for illustration, we speculate that a change of bag colour from grey to red, green or blue in the city of Paris could be equivalent to installing up to 8,370, 9,537 or 13,600 additional bins, respectively.”

Which is admissible in my opinion. However, In the abstract the sentence at lines 20 – 22 the sentence is formulated in a different way, leading to the impression that this is a result of the study.

“..assuming a linear increase in detection, we illustrate that this change could be equivalent to installing up to 8,370 bins in the city of Paris, in terms of perceived density.”

Also at lines 106 – 110:

“We then convert the effect to a measure of perceived bin density and approximate the equivalent number of real bins that would have to be added in the streets of Paris to achieve the same effect..”

The calculation of the “the equivalent number of real bins” still puzzles me, because in some contexts it is presented as a result of the study when in my opinion it is speculative reasoning. The linear function for the conversion here is just assumed (as you state at line 410). There is no evidence for the fact that this relationship should be linear, not even support from previous literature. Changing the colour of the bag as demonstrated in your study increases bin detection when the observer 1) is actively looking for the bin in the images 2) the image is presented for a limited amount of time. In most real-life situations, the time to visually scan the environment is not limited to a few milliseconds. When time pressure is low or absent, the advantage given by colour (or another feature that attracts attention in a bottom-up manner) will be smaller much smaller than then what found in your study. The speed accuracy trade-off in visual search is a well-known phenomenon, with reduced time pressure the accuracy will increase also for the hard serial scanning and the advantage given by bottom-up cues will be reduced. The increase in single bin detection you found in your images cannot therefore be linearly translated into perceived bin density in an ecological context where the time constrain is much less stringent. Consequently, the equivalence between the increase in detection at your task and the number of bins installed under ecological conditions is purely speculative. Unless you have a way to substantiate (with data or citations) the existence of a linear relationship between single bin detection in your experiment and perceived bin density under real-life conditions, I would ask you to rephrase or remove from the text all sentences (including those in the abstract) related to the claim of equivalence between perceived density and the number of bins that would need to be installed to achieve a similar effect such as on line 313 (similar to the addition of 13,600 new bins).

Reviewer #4: This is an interesting article that merits publication, provided appropriate changes are made. In particular, additional details are needed regarding the participants, as well as the data analysis and findings. Furthermore, some of the conclusions go beyond what can be supported by the findings.

More information should be provided regarding the participants who were included and excluded. Participants were retained if they responded in at least 70% of the trials (and met other inclusion criteria), but a 30% missing data rate seems quite high. What was the average rate of missing data? Could an argument be made for treating the trials with no response as misses (when a trash bag was present) or correct rejections (when no bag was present)? How might this have affected the d' values?

Studies 1 and 2 had far more females than males, while Study 4 showed the opposite trend. Were these gender imbalances the result of excluding more participants from one gender than the other, or did they simply reflect differences in the rates at which the two genders volunteered to participate?

The authors chose not to exclude participants who described themselves as having color blindness (which should be described in the paper as "anomalous color vision). They claimed this would increase the generalizability of the findings, but males are much more likely to have anomalous color vision than females. Combining this with the gender imbalance of the studies means that ignoring gender and self-reported anomalous color vision in the analyses can actually reduce generalizability. The solution would be to include self-reported anomalous color vision as a covariate (although many people are not aware they have anomalous color vision), or testing the data for gender differences.

The manuscript provides very few descriptive statistics. Perhaps these could be provided in the OSF repository. In particular, hit and false alarm rates for each condition should be reported. Furthermore, d' values can only be calculated when hit and false alarm rates are greater than 0% and less than 100%. Were rates of 0% or 100% obtained for some participants in some conditions? If so, how often? What corrections were used to calculate d'?

In Study 1, the false alarm rate for calculating the grey bag d' is the same as the false alarm rate for calculating the red bag d'. Because of this, when the paired t-test was conducted, the false alarm rate drops out and the t-test is based entirely on the grey and red hit rates:

d' difference = z(Red hit rate) - z(FA) - [z(Grey hit rate) - z(FA)] = z(Red hit rate) - z(Grey hit rate)

This points to one of the issues in extrapolating from the d' values to the equivalent number of trash bins: The false alarm rates are ignored entirely in that calculation.

Furthermore, d' is a nonlinear variable that is difficult for most people (including policy makers) to understand. There is certainly tremendous appeal in an alternative metric that is more easily understood, such as the equivalent number of trash bins. However, it is a huge leap to jump from changes in d' to equivalent numbers of trash bins (and to jump from increases in the number of trash bins to decreases in littering, as previous reviewers have noted). It might be better to convert the d' values into estimated areas under the received operating characteristic (ROC). Think of the ROC area in relation to a task in which participants see two photographs. One contains a trash bag (of the color under study) and the other does not. The participant's task is to identify the picture that contains the trash bag. If the trash bag is easy to see (as in the catch trials) performance will be 100%. If the trash bag is virtually impossible to see the participant must guess and performance will be at the level of chance: 50%. To convert d' into RC area estimates, use the formula provided by Macmillan (1993) in Keren and Lewis (Eds.), A handbook for data analysis in the behavioral sciences: Methodological issues:

ROC area = inverse-z[ d' ÷ sqrt(2) ]

This formula predicts an ROC area of 87.8% for the red bags in Study 1 and 82.8% for the grey bags. Both bags have reasonably good saliency; performance is well above the chance level of 50%. Performance is 6% better with the red bag than with the grey bag. That may indeed be a notable increase in saliency, but it's a major leap to conclude that this is equivalent to a 28% increase in the number of trash bins. That simply wasn't tested.

Study 3 seems very much out of place and isn't even mentioned in the Abstract. If it is retained (which I do not recommend) it needs to cite literature on how policy makers reach decisions -- and there needs to be a discussion explaining why the results from the participants (who are probably not policy makers by profession) can be generalized to actual policy makers.

The authors have responded to many of the comments made by previous reviewers. However, there is surprisingly little mention of the need to conduct a field study to validate the findings. It's one thing to present photographs on a computer screen to participants who are instructed to find trash bins in those photographs; it may be entirely different to have people in natural settings decide what they're going to do with their litter. The studies reported by the authors are a good start, but there needs to be better recognition of their limitations.

One minor point concerns the descriptions of the participants in Study 2. The second set of statistics presumably applies to Study 2.B, but this is not indicated in the text.

7. PLOS authors have the option to publish the peer review history of their article (what does this mean?). If published, this will include your full peer review and any attached files.

Reviewer #1: No

Reviewer #2: No

Reviewer #3: No

Reviewer #4: No

---

## [Author Response · Author response to Decision Letter 1]

31 May 2022

Dear editor and reviewers, please refer to the uploaded "Response to reviewers" document for a more appropriate format of equations and tables.

---

## [Decision Letter · Decision Letter 2]

1 Sep 2022

PONE-D-21-08195R2A zero-cost attention-based approach to promote cleaner streets: a Signal Detection Theory approach in Parisian streetsPLOS ONE

Dear Dr. Abdel Sater,

thank you again for submitting your revised manuscript to PLOS ONE. After careful consideration by the Reviewers and given their reports, we feel that your revision substantially improved the manuscript but it still does not fully meet PLOS ONE’s publication criteria as it currently stands.  We invite you therefore to submit a revised version of your work that addresses the points raised by Reviewers.

In your revised version, please make sure to carefully address each point raised in Reviewers' comments, especially, concerning the statistical and technical aspects of your work as mentioned in comments of Reviewer 2 below.

Please submit your revised manuscript within six months fom this date as afterwards, any revision has to be considered as a new submission. If you will need more time than this to complete your revisions, please reply to this message or contact the journal office at plosone@plos.org. Please include the following items when submitting your revised manuscript:A rebuttal letter that responds to each point raised by the academic editor and reviewer(s). You should upload this letter as a separate file labeled 'Response to Reviewers'.A marked-up copy of your manuscript that highlights changes made to the original version. You should upload this as a separate file labeled 'Revised Manuscript with Track Changes'.An unmarked version of your revised paper without tracked changes. You should upload this as a separate file labeled 'Manuscript'.

Thank you for considering PLOS ONE for reporting your research.  We look forward to receiving your revised manuscript.

Kind regards,

Sasha

Alexander N. 'Sasha' Sokolov, Ph.D.

Academic Editor

PLOS ONE

Reviewers' comments:

Reviewer's Responses to Questions

**Comments to the Author**

1. If the authors have adequately addressed your comments raised in a previous round of review and you feel that this manuscript is now acceptable for publication, you may indicate that here to bypass the “Comments to the Author” section, enter your conflict of interest statement in the “Confidential to Editor” section, and submit your "Accept" recommendation.

Reviewer #2: (No Response)

Reviewer #3: All comments have been addressed

Reviewer #4: (No Response)

2. Is the manuscript technically sound, and do the data support the conclusions?

Reviewer #2: Yes

Reviewer #3: Yes

Reviewer #4: Yes

3. Has the statistical analysis been performed appropriately and rigorously? 

Reviewer #2: No

Reviewer #3: Yes

Reviewer #4: Yes

4. Have the authors made all data underlying the findings in their manuscript fully available?

Reviewer #2: Yes

Reviewer #3: Yes

Reviewer #4: Yes

5. Is the manuscript presented in an intelligible fashion and written in standard English?

Reviewer #2: Yes

Reviewer #3: Yes

Reviewer #4: Yes

6. Review Comments to the Author

Reviewer #2: The authors have put considerable effort into revising the manuscript “A zero-cost attention-based approach to promote cleaner streets: a Signal Detection Theory approach in Parisian streets” and addressing my previous comments and concerns. The conducted research is now reported in a more concise and clear manner, and elaboration around relevant background literature (e.g. nudging) as well as study limitations have been included. Therefore, the rationale as well as practical implications of the research are now presented in a more sound manner, and overall the authors have made a stronger case for the relevance and significance of their research to the field of environmental psychology. However, some questions still remain surrounding the statistics, and some minor points regarding written expression have been listed in my comments below. Therefore I have recommended minor revisions.

Whilst this has improved significantly, the main area that still needs work is the reporting of the statistics. First, there is confusion on the chosen post-hoc analysis for Study 2a and b (row 330 onwards). From my understanding, after doing a 3x2 ANOVA, they examined the interaction further by running 3 separate 2x2 ANOVAS. I’m not convinced this has achieved their goal and still leaves questions on where the significant interaction lies. I’d recommend doing Simple Effects Analysis, e.g. comparing the three studies for the grey condition – so not statistically testing an interaction, but looking at a main effect of study but just on the grey condition (based on Fig 4, I suspect there will not be any differences) and then again for the 3 colours (of which I suspect some clear effects). This would lead to a much clearer and simpler narration of your findings (i.e. the studies found participants responded the same for the grey conditions, but there was a difference in colour…). The initial ANOVA was also not fully reported. As well as reporting the interaction, it is important to also report the main effects (e.g. study and colour). Second, the reporting of the ANOVA for Study 3 is unclear and needs further revision. E.g. the interaction and main effects need to be spelled out fully (row 381-382).

Third and finally, the effect sizes need to be checked. This recent addition does make the paper stronger and clearer for the reader to follow, but the values are questionable. In the footnote, it implies Cohen’s dz is being used yet the text just notes d (and not dz). This needs to be explicit. For example, why is Cohen’s d not appropriate here (of which, should be under the value of 1)? If sticking with Cohen’s dz, the results are incorrect. I copied the footnotes’ formula, but did not replicate all of the reported findings. This needs to be checked.

In terms of minor points:

- Abstract, row 17: would be useful to clarify “trash can detection rates”.

- Row 56 & 60 -> unsupported claim needing a reference (even is this is not publically available, a personal correspondence reference is needed).

- row 65 ->: The authors write: “we ask whether an increase in the visual saliency of bins can work as a substitute to adding bins”, but they have not compared these two strategies. I suggest rewording this part for clarity.

- row 68 ->: references to “a number of studies” are needed.

- Row 189 -> unsupported claim needing a reference

- Row 199 -> the numbers still do not add up (203 females + 104 males = 307, not 308). As noted in the previous review, if there is a respondent who didn’t state their gender, this needs to be explicit.

- Row 259-260 -> unsupported claim needing a reference

- row 262 ->: explanation of how ROC was estimated and why could go in the Method section.

- Row 297 -> could state “As applied in Study 1, trails for which…” to emphasise continuity between studies.

- Rows 318-319 -> be consistent in decimal places (e.g. should the SDs have .00 added to them to be in-line with the M units?)

- row 350: should read “Study 3”.

- Row 351 -> “replicated the study” – this is confusing as three studies have previously been reported. This could be rephrased to “replicated the design as used in the previous studies…”

- Throughout -> as above, the manuscript often refers to “the Study” (e.g. row 65 & 396) yet 3 studies are reported. Rephrase (e.g. “this paper”?)

- Throughout -> ensure spelling is consistent (E.g. US spelling – color; or UK spelling – colour)

- Throughout -> check formatting (e.g. unnecessary spacing row 311, missing full stops row 385, paragraph spacing)

Reviewer #3: I believe that the manuscript in its current form reflects the standards of PLOS ONE, and I recommend its acceptance.

Reviewer #4: This submission addresses the concerns I had regarding previous versions. I have only a few suggested changes, all of them minor.

The participants were asked whether or not they had colour blindness, but the proper term is anomalous color vision. I would suggest using the term anomalous colour vision, and explaining (in parentheses) that participants were asked if they had colour blindness. This applies to Lines 203 and 302, and the footnote on page 15.

In Lines 267-268, readers might confuse the explanatory statistics (ROC area of 0.6) with the study findings (ROC areas of 87.8% and 82.8%). I suggest rephrasing the text using the actual study findings. For example, for the red bag the ROC area of .878 indicates that participants would were 87.8% accurate in identifying the presence of red bags.

There are a few minor mathematical inconsistencies that should be addressed (unless they are due to rounding issues). On Line 268, the two ROC areas differ by 5% rather than 6%. On Line 376, the d’ for blue bags appears to be 40% higher than for grey, not 39%. On Line 383, the ROC areas appear to be 9.0% higher for blue than for grey, not 8.9%.

In the Discussion, very briefly discuss the possibility of policy makers having concerns about increasing the saliency of the bags, especially in areas of Paris that target tourists. It may be that a conscious decision was made to select an “invisible” bag color so as not detract from the visual surroundings.

In the Supplementary material, add information about the number of participants excluded for each reason in each study.

Finally, there are some minor typographic corrections. In Line 204, “of the declared” should read “of them declared.” On Line 262, delete the quotation at start of the sentence. On Line 255, round the mean to 1114 (to maintain same level of precision as the mean reaction time for red bags).

7. PLOS authors have the option to publish the peer review history of their article (what does this mean?). If published, this will include your full peer review and any attached files.

Reviewer #2: No

Reviewer #3: **Yes: **Giulio Contemori

Reviewer #4: No

---

## [Author Response · Author response to Decision Letter 2]

17 Nov 2022

Reviewers' comments

Dear editor,

Thank you for giving us the opportunity to submit a revised draft of our manuscript. We appreciate the time and effort that you and the reviewers have dedicated to providing your feedback on our manuscript. We have incorporated the reviewers’ suggestions and highlighted the changes in red within the manuscript.

 You will find below a point-by-point response to the reviewers’ comments.

Reviewer #2: The authors have put considerable effort into revising the manuscript “A zero-cost attention-based approach to promote cleaner streets: a Signal Detection Theory approach in Parisian streets” and addressing my previous comments and concerns. The conducted research is now reported in a more concise and clear manner, and elaboration around relevant background literature (e.g. nudging) as well as study limitations have been included. Therefore, the rationale as well as practical implications of the research are now presented in a more sound manner, and overall the authors have made a stronger case for the relevance and significance of their research to the field of environmental psychology. However, some questions still remain surrounding the statistics, and some minor points regarding written expression have been listed in my comments below. Therefore I have recommended minor revisions.

Whilst this has improved significantly, the main area that still needs work is the reporting of the statistics. First, there is confusion on the chosen post-hoc analysis for Study 2a and b (row 330 onwards). From my understanding, after doing a 3x2 ANOVA, they examined the interaction further by running 3 separate 2x2 ANOVAS. I’m not convinced this has achieved their goal and still leaves questions on where the significant interaction lies. I’d recommend doing Simple Effects Analysis, e.g. comparing the three studies for the grey condition – so not statistically testing an interaction, but looking at a main effect of study but just on the grey condition (based on Fig 4, I suspect there will not be any differences) and then again for the 3 colours (of which I suspect some clear effects). This would lead to a much clearer and simpler narration of your findings (i.e. the studies found participants responded the same for the grey conditions, but there was a difference in colour…). 

We agree that adding a simple effect analysis clarifies the nature of the interaction. The results of the simple effects analysis were thus added in the results section:

“An analysis of simple effects showed that the treatment effect was significant for the colour condition (between the different colours), F (2,912) = 6.40, p < 0.01, but not in the grey condition F (2, 912) = 0.73, p = 0.48. This suggests that participants responded the same for the grey condition across the three studies, but there was a difference in the coloured trial conditions.”

However, we maintain the original two by two analysis because it is important to demonstrate that some colours are more superior compared to grey than others.

The initial ANOVA was also not fully reported. As well as reporting the interaction, it is important to also report the main effects (e.g. study and colour). 

We thank the reviewer for pointing this out. The main effects were added to the results of the mixed ANOVA in the text:

“The main effect for COLOUR was significant, F (1, 912) = 1191.40, p < .01, η^2= .57, but the he main effect for STUDY was not significant, F (2, 912) = 2.73, p = .07, η^2= .01.”

Second, the reporting of the ANOVA for Study 3 is unclear and needs further revision. E.g. the interaction and main effects need to be spelled out fully (row 381-382).

We agree with the reviewer that the ANOVA reporting was unclear in the text. We added the following:

“A mixed model ANOVA testing 2-level COLOUR factor (grey vs colour) x 2-level STUDY factor (study 2.B vs study 3) confirms that the effect of changing the colour from grey to blue was greater when considering a sample of residents in the Paris Area compared to a British sample; the interaction between STUDY and COLOUR was significant F(1,589) = 14.91, p < .001, η^2 = .025. The main effects were also significant; STUDY F(1,588) = 5.07, p = .02, η^2 = .01, COLOUR F(1,588) = 1585.63, p < .00, η^2 = .73.” 

Third and finally, the effect sizes need to be checked. This recent addition does make the paper stronger and clearer for the reader to follow, but the values are questionable. In the footnote, it implies Cohen’s dz is being used yet the text just notes d (and not dz). This needs to be explicit. For example, why is Cohen’s d not appropriate here (of which, should be under the value of 1)? If sticking with Cohen’s dz, the results are incorrect. I copied the footnotes’ formula, but did not replicate all of the reported findings. This needs to be checked.

When comparing dependent means, the correlation between the observations has to be taken into account, and the effect size directly related to the statistical significance of the test (and thus used in power analysis) is Cohen’s dz (see Lakens, 2013). We agree that this should have been stated more clearly. 

Footnote 2 was changed to:

“The standardized mean difference effect size for within-subjects designs is referred to as Cohen's dz and its formula is based on calculations by Rosenthal (1991):d_z= t/√n, , given the direct relationship between the t-value of a paired-samples t-test and Cohen's dz”

Similarly, all instances of “d” were changed to “dz.

Lakens, D. (2013). Calculating and reporting effect sizes to facilitate cumulative science: a practical primer for t-tests and ANOVAs. Frontiers in Psychology, 4, 1-12. [863].

In terms of minor points: 

- Abstract, row 17: would be useful to clarify “trash can detection rates”.

This sentence has been changed.

- Row 56 & 60 -> unsupported claim needing a reference (even is this is not publically available, a personal correspondence reference is needed).

The following reference has been added: 

16. France3 Paris. Paris : nouveau design pour les poubelles parisiennes. France 3 Paris Ile-de-France [Internet]. 2013 Nov [cited 3 nov 2022]. Available from: https://france3-regions.francetvinfo.fr/paris-ile-de-france/paris/paris-nouveau-design-pour-les-poubelles-parisiennes-366167.html

- row 65 ->: The authors write: “we ask whether an increase in the visual saliency of bins can work as a substitute to adding bins”, but they have not compared these two strategies. I suggest rewording this part for clarity.

We agree with the reviewer that this sentence was not very clear. It has thus been changed to: 

“In this study, we ask whether an increase in the visual saliency of bins can increase the perceived density of bins in a city.”

- row 68 ->: references to “a number of studies” are needed.

The following references were added to that sentence: 

19. Arnold M. The relationship between receptacle design, normative conduct, environmental concerns, and recycling behavior. :37. 

23 Geller ES, Brasted WS, Mann MF. Waste Receptacle Designs as Interventions for Litter Control. Journal of Environmental Systems. 1 janv 1979;9(2):145‑60. 

24. O’Neill GW, Blanck LS, Joyner MA. The use of stimulus control over littering in a natural setting. J Appl Behav Anal. 1980;13(2):379‑81

- Row 189 -> unsupported claim needing a reference

The following reference was added: 

35. Afrin S, Rahman A, Islam F, Hoque F. Environmental effects of tourism. 2013;1(7):15. 

- Row 199 -> the numbers still do not add up (203 females + 104 males = 307, not 308). As noted in the previous review, if there is a respondent who didn’t state their gender, this needs to be explicit.

This typo was corrected.

- Row 259-260 -> unsupported claim needing a reference

The following reference was added: 

43. Wong AL, Haith AM, Krakauer JW. Motor planning. Neuroscientist 21: 385–398, 2015. doi:10.1177/1073858414541484.

- row 262 ->: explanation of how ROC was estimated and why could go in the Method section.

We thank the reviewer for this suggestion. The following sentence was added at the end of the methods section: 

“Then, to measure the change in performance in a more intuitive manner, we estimated the area under the received operating characteristic (ROC) of each colour condition using the d^' as follows (Macmillan, 1993) (38): A_(d^' )= ϕ(d^'/√2), where Φ(∙) corresponds to the cumulative normal distribution. The ROC area can be interpreted as the proportion of times subjects would correctly identify the signal (39). For example, if the trash bag is easy to see, ROC area will be equal to 1. If the trash bag is virtually impossible to detect the participant must guess and the ROC area will be at level of chance: 0.5.”

- Row 297 -> could state “As applied in Study 1, trails for which…” to emphasise continuity between studies.

- Rows 318-319 -> be consistent in decimal places (e.g. should the SDs have .00 added to them to be in-line with the M units?)

- Row 351 -> “replicated the study” – this is confusing as three studies have previously been reported. This could be rephrased to “replicated the design as used in the previous studies…”

- Throughout -> as above, the manuscript often refers to “the Study” (e.g. row 65 & 396) yet 3 studies are reported. Rephrase (e.g. “this paper”?)

- Throughout -> ensure spelling is consistent (E.g. US spelling – color; or UK spelling – colour)

All of these revisions have been addressed. 

Reviewer #4: This submission addresses the concerns I had regarding previous versions. I have only a few suggested changes, all of them minor.

The participants were asked whether or not they had colour blindness, but the proper term is anomalous color vision. I would suggest using the term anomalous colour vision, and explaining (in parentheses) that participants were asked if they had colour blindness. This applies to Lines 203 and 302, and the footnote on page 15.

The instances of “colour blindness” in the text was changed to “anomalous colour vision” as suggested. 

In Lines 267-268, readers might confuse the explanatory statistics (ROC area of 0.6) with the study findings (ROC areas of 87.8% and 82.8%). I suggest rephrasing the text using the actual study findings. For example, for the red bag the ROC area of .878 indicates that participants would were 87.8% accurate in identifying the presence of red bags.

We agree that the wording was confusing to the reader. The sentence now reads: 

“ROC area calculations show that participants were accurately identified the bin in 87.8% of the trials (A_(d_red^' )= 0.878) and 82.8% accurate in identifying the presence of a grey bag (A_(d_grey^' )= 0.828). Participants were thus 5% more likely to correctly identify a bin with the red bag compared to the grey one”. 

There are a few minor mathematical inconsistencies that should be addressed (unless they are due to rounding issues). On Line 268, the two ROC areas differ by 5% rather than 6%. On Line 376, the d’ for blue bags appears to be 40% higher than for grey, not 39%. On Line 383, the ROC areas appear to be 9.0% higher for blue than for grey, not 8.9%.

The rounding inconsistency was corrected.

In the Discussion, very briefly discuss the possibility of policy makers having concerns about increasing the saliency of the bags, especially in areas of Paris that target tourists. It may be that a conscious decision was made to select an “invisible” bag color so as not detract from the visual surroundings.

The following paragraph was added in the discussion. 

“While increasing the saliency of bins can decrease littering, it might entail esthetic costs. This is especially problematic in touristic cities with architectural heritage, like Paris. To help policy makers make informed decisions facing this trade-off, more evidence on the benefits of more visible bins is needed”.

In the Supplementary material, add information about the number of participants excluded for each reason in each study.

We thank the reviewer, and we agree that this information needs to be provided. We add the following table to the Supplementary materials. 

Table 6 Participant exclusion rates

 Initial N Failed 5 or more catch trials Unrealistic Reaction time on more than 30% of trials Final N

Study 1 (red) 324 16 0 307

Study 2.A (green) 312 11 0 301

Study 2.B (blue) 315 7 0 308

Study 3 (blue – French sample) 307 24 1 282

Finally, there are some minor typographic corrections. In Line 204, “of the declared” should read “of them declared.” On Line 255, round the mean to 1114 (to maintain same level of precision as the mean reaction time for red bags).

This typo was corrected

---

## [Decision Letter · Decision Letter 3]

24 Jan 2023

PONE-D-21-08195R3A zero-cost attention-based approach to promote cleaner streets: a Signal Detection Theory approach in Parisian streetsPLOS ONE

Dear Dr. Abdel Sater,

thank you for submitting your revised manuscript to PLOS ONE.  After careful consideration by two Reviewers who both have mentioned a considerable improvement of the manuscript, we feel that some additional work is still required for your paper to fully meet PLOS ONE’s publication criteria.  Therefore, we invite you to submit a revised version of the manuscript that addresses the points raised, in particular, by Reviewer 2 (see below) regarding statistical and technical aspects of your work. 

Please submit your revised manuscript within six months from this date as thereafter, any revision has to be considered a new submission.  If you will need more time than this to complete your revisions, please reply to this message or contact the journal office at plosone@plos.org. Please include the following items when submitting your revised manuscript:A rebuttal letter that responds to each point raised by the academic editor and reviewer(s). You should upload this letter as a separate file labeled 'Response to Reviewers'.A marked-up copy of your manuscript that highlights changes made to the original version. You should upload this as a separate file labeled 'Revised Manuscript with Track Changes'.An unmarked version of your revised paper without tracked changes. You should upload this as a separate file labeled 'Manuscript'.

We look forward to receiving your revised manuscript.

Thank you for choosing PLOS ONE for reporting your research.

Kind regards,

Sasha

Alexander N. 'Sasha' Sokolov, Ph.D.

Academic Editor

PLOS ONE

Journal Requirements:

Reviewers' comments:

Reviewer's Responses to Questions

**Comments to the Author**

1. If the authors have adequately addressed your comments raised in a previous round of review and you feel that this manuscript is now acceptable for publication, you may indicate that here to bypass the “Comments to the Author” section, enter your conflict of interest statement in the “Confidential to Editor” section, and submit your "Accept" recommendation.

Reviewer #2: (No Response)

Reviewer #4: (No Response)

2. Is the manuscript technically sound, and do the data support the conclusions?

Reviewer #2: Yes

Reviewer #4: Yes

3. Has the statistical analysis been performed appropriately and rigorously? 

Reviewer #2: Yes

Reviewer #4: Yes

4. Have the authors made all data underlying the findings in their manuscript fully available?

Reviewer #2: Yes

Reviewer #4: No

5. Is the manuscript presented in an intelligible fashion and written in standard English?

Reviewer #2: Yes

Reviewer #4: Yes

6. Review Comments to the Author

Reviewer #2: The authors have revised their manuscript “A zero-cost attention-based approach to promote cleaner streets: a Signal Detection Theory approach in Parisian streets” as per my previous comments. My comments regarding the reported statistics have now been addressed, and the authors have provided clarifications to some of the issues I raised previously. I have a few additional suggestions regarding the results reporting as well as a few points regarding written expression. After these have been addressed, I feel this is appropriate to be accepted for publication. Well done.

The minor points (row numbers refer to the tracked changes document):

Introduction:

- rows 47 – 48 : “increase in the number of bins might increase litter in the presence of litter in the environment” -> Rewording suggested for clarity.

- Rows 95-97: update to a more relevant example or remove

All Method sections

- Provide a brief justification of the target sample size. Each study had a very specific sample size, considering the researchers paid for this amount, an explicit rationale is needed in why they asked Prolific for each sample size

All Results sections

- Ensure formatting of the results is consistent throughout the paper (e.g. the letters should be italicised, but the numbers not, p value should be reported fully unless < .001 [row 335+]; all decimals should be the same e.g. to 2 decimal places)

- Provide a recap summary of the main take home finding(s) for each study before moving on to the next section. Just needs to be 1-2 sentences.

Methods (Study 1):

- rows 258 – 259: “The difference between mean discriminability in the two colour conditions” should be rephrased as “The difference in mean discriminability between the two colour conditions” for clarity.

Methods (Study 2):

- rows 324-327: statements need to be supported with reference(s).

Results (Study 2):

- row 371 onwards: When reporting the results from the mixed model ANOVA, the dependent variable should be stated.

- rows 399 – 401: I would suggest moving this part about the Study 2 -specific ROC results before the mixed ANOVA. In this way all Study 2 -specific results are grouped, ensuring a more logical flow.

- row 425: can omit “also”.

Methods (Study 3):

- No colour blindness rates reported for this sample. Could state that this data could not be collected.

Results (Study 3):

- e.g row 436 onwards: Should clearly state what the dependent variable was, e.g. “… the effect of changing the colour from grey to blue on d’ was greater when…”.

- row 471 onwards: “increasing the saliency of bins can decrease littering might entail esthetic costs…” should be reworded for clarity (also esthetic -> aesthetic).

- rows 494 – 495: “In all these cases, the bin is at the same distance in the eye-catching condition and in the control condition but littering rates are different.” -> Not clear whether this refers to Studies 1 – 3 or to the existing studies reviewed just before.

- row 508: Should be “colours”.

- row 508: Omit “Therefore” to avoid repetition.

Reviewer #4: The use of italics on lines 338 and 348-351 differs from the use of italics elsewhere in the manuscript. Also, line 429 is missing a word -- or perhaps "can" should be replaced with "to."

7. PLOS authors have the option to publish the peer review history of their article (what does this mean?). If published, this will include your full peer review and any attached files.

Reviewer #2: No

Reviewer #4: No

---

## [Author Response · Author response to Decision Letter 3]

13 Feb 2023

We thank the reviewers for examining the article in detail and for providing insightful comments. We have addressed all the minor points pointed out in this review. Please see details below.

The minor points (row numbers refer to the tracked changes document):

Introduction:

- rows 47 – 48 : “increase in the number of bins might increase litter in the presence of litter in the environment” -> Rewording suggested for clarity.

Done

- Rows 95-97: update to a more relevant example or remove

Done

All Method sections

- Provide a brief justification of the target sample size. Each study had a very specific sample size, considering the researchers paid for this amount, an explicit rationale is needed in why they asked Prolific for each sample size

Done

All Results sections

- Ensure formatting of the results is consistent throughout the paper (e.g. the letters should be italicised, but the numbers not, p value should be reported fully unless < .001 [row 335+]; all decimals should be the same e.g. to 2 decimal places)

Done

- Provide a recap summary of the main take home finding(s) for each study before moving on to the next section. Just needs to be 1-2 sentences.

We thank the reviewer for this suggestion. We have added the following summaries after each results section: 

“In summary, bins with red bags are more visible than bins with grey bags.”

“In summary, Study 2’s results of show that changing the colour of bags from grey to green and from grey to blue significantly increases their visibility. A comparison between the three colour conditions shows that a bin with a blue bag is significantly more visible than one with red or green bag.”

“In summary, we observe the same results when we consider a Parisian (local) sample; the visibility of a bin increases when its bag is blue compared to grey. Study 3 also shows that the change in visibility is greater for Parisian participants than it is for British participants.”

Methods (Study 1): 

- rows 258 – 259: “The difference between mean discriminability in the two colour conditions” should be rephrased as “The difference in mean discriminability between the two colour conditions” for clarity.

Done

Methods (Study 2):

- rows 324-327: statements need to be supported with reference(s).

Since this observation resulted from personal exchanges with employees of the Parisian munucupality, the following reference has been added: 

“(Personal communication, the department of cleanliness and sanitation (DPE) of the Paris municipality, 2021).”

Results (Study 2):

- row 371 onwards: When reporting the results from the mixed model ANOVA, the dependent variable should be stated.

Done

- rows 399 – 401: I would suggest moving this part about the Study 2 -specific ROC results before the mixed ANOVA. In this way all Study 2 -specific results are grouped, ensuring a more logical flow.

Done

- row 425: can omit “also”.

Done

Methods (Study 3):

- No colour blindness rates reported for this sample. Could state that this data could not be collected.

We thank the reviewer for pointing this out. The following sentence has been added: 

“2% of the sample reported anomalous colour vision (participants were asked whether they were colour blind) and 61% of them declared wearing glasses or contact lenses to correct their vision.”

Results (Study 3):

- e.g row 436 onwards: Should clearly state what the dependent variable was, e.g. “… the effect of changing the colour from grey to blue on d’ was greater when…”.

Done

- row 471 onwards: “increasing the saliency of bins can decrease littering might entail esthetic costs…” should be reworded for clarity (also esthetic -> aesthetic).

Done

- rows 494 – 495: “In all these cases, the bin is at the same distance in the eye-catching condition and in the control condition but littering rates are different.” -> Not clear whether this refers to Studies 1 – 3 or to the existing studies reviewed just before.

Done

- row 508: Should be “colours”.

Done

- row 508: Omit “Therefore” to avoid repetition.

Done

Reviewer #4: The use of italics on lines 338 and 348-351 differs from the use of italics elsewhere in the manuscript. Also, line 429 is missing a word -- or perhaps "can" should be replaced with "to."

Done

---

## [Decision Letter · Decision Letter 4]

28 Mar 2023

A zero-cost attention-based approach to promote cleaner streets: a Signal Detection Theory approach in Parisian streets

PONE-D-21-08195R4

Dear Dr. Abdel Sater,

thank you for your efforts with the final round of revision.  We’re pleased to inform you that your manuscript has now been judged scientifically suitable for publication and will be formally accepted for publication once it meets all outstanding technical requirements.

Thank you again for choosing PLOS ONE for reporting your work. 

Kind regards,

Sasha

Alexander N. 'Sasha' Sokolov, Ph.D.

Academic Editor

PLOS ONE

Additional Editor Comments (optional):

Reviewers' comments:

Reviewer's Responses to Questions

**Comments to the Author**

1. If the authors have adequately addressed your comments raised in a previous round of review and you feel that this manuscript is now acceptable for publication, you may indicate that here to bypass the “Comments to the Author” section, enter your conflict of interest statement in the “Confidential to Editor” section, and submit your "Accept" recommendation.

Reviewer #2: All comments have been addressed

2. Is the manuscript technically sound, and do the data support the conclusions?

Reviewer #2: Yes

3. Has the statistical analysis been performed appropriately and rigorously? 

Reviewer #2: Yes

4. Have the authors made all data underlying the findings in their manuscript fully available?

Reviewer #2: No

5. Is the manuscript presented in an intelligible fashion and written in standard English?

Reviewer #2: Yes

6. Review Comments to the Author

Reviewer #2: Overall, this paper has evolved nicely through the revisions. All previous proposed revisions have now been addressed and I am happy for it to be accepted. I did notice some grammatical mistakes following the latest revisions (e.g. the sentence "... using outcome variances computed in the pilot study over 100 participants" doesn't follow and may be missing a word; the summary statements should be in past tense; "can might" within "is important to note that increasing the saliency of bins can might entail aesthetic costs") however the next stage should capture these.

Well done team, its a lovely paper

7. PLOS authors have the option to publish the peer review history of their article (what does this mean?). If published, this will include your full peer review and any attached files.

Reviewer #2: No

---

## [Editor Report · Acceptance letter]

3 Apr 2023

PONE-D-21-08195R4 

A zero-cost attention-based approach to promote cleaner streets: a Signal Detection Theory approach in Parisian streets 

Dear Dr. Abdel Sater:

I'm pleased to inform you that your manuscript has been deemed suitable for publication in PLOS ONE. Congratulations! Your manuscript is now with our production department. 

Kind regards, 

on behalf of

Dr. Alexander N. Sokolov 

Academic Editor

PLOS ONE